# Blockchain-Enabled Infection Sample Collection System Using Two-Echelon Drone-Assisted Mechanism

Shengqi Kang [1] and Xiuwen Fu [2,*]

[1] Institute of Logistics Science and Engineering, Shanghai Maritime University, Shanghai 201306, China; kangshengqi@stu.shmtu.edu.cn
[2] Logistics Engineering College, Shanghai Maritime University, Shanghai 201306, China
* Correspondence: xwfu@shmtu.edu.cn

**Abstract:** The collection and transportation of samples are crucial steps in stopping the initial spread of infectious diseases. This process demands high levels of safety and timeliness. The rapid advancement of technologies such as the Internet of Things (IoT) and blockchain offers a viable solution to this challenge. To this end, we propose a Blockchain-enabled Infection Sample Collection system (BISC) consisting of a two-echelon drone-assisted mechanism. The system utilizes collector drones to gather samples from user points and transport them to designated transit points, while deliverer drones convey the packaged samples from transit points to testing centers. We formulate the described problem as a Two-Echelon Heterogeneous Drone Routing Problem with Transit point Synchronization (2E-HDRP-TS). To obtain near-optimal solutions to 2E-HDRP-TS, we introduce a multi-objective Adaptive Large Neighborhood Search algorithm for Drone Routing (ALNS-RD). The algorithm's multi-objective functions are designed to minimize the total collection time of infection samples and the exposure index. In addition to traditional search operators, ALNS-RD incorporates two new search operators based on flight distance and exposure index to enhance solution efficiency and safety. Through a comparison with benchmark algorithms such as NSGA-II and MOLNS, the effectiveness and efficiency of the proposed ALNS-RD algorithm are validated, demonstrating its superior performance across all five instances with diverse complexity levels.

**Keywords:** blockchain; drone-assisted sample collection; routing; adaptive large neighborhood search; synchronization





## 1. Introduction

Infectious diseases are a class of diseases that can be transmitted between humans, animals, or between humans and animals. They are caused by various pathogens, which pose a serious threat to human life and property security. One example is the outbreak of the COVID-19 pandemic, which has had huge impacts on people's health, the global economy, and the international community [1]. As of March 2023, there have been more than 470 million confirmed cases and over 6 million deaths worldwide [2]. As a result of COVID-19, economic growth has declined and unemployment has risen globally in recent years [3]. According to a United Nations report, the global economic growth is expected to be 1.9% in 2023, one of the lowest growth rates in decades.

Infectious diseases like COVID-19 exhibit strong infectivity and long incubation periods, leading to widespread transmission and difficulties in controlling the spread. Delayed detection of outbreaks has resulted in city-wide lockdowns in many countries. For urgent public health emergencies, the Centers for Disease Control and Prevention (CDC) must ensure continuous and sensitive monitoring for early warning, which is beneficial in preventing virus spread and disease transmission [4]. The CDC typically conducts regular sample testing in high-risk communities during the initial stages of an outbreak. Sample collection and transportation are crucial links that require high levels of safety and timeliness.

When collecting samples within the community, residents traditionally undergo testing at hospitals or community sites, a process known for its lengthy sampling times and the potential for gatherings that contribute to the spread of viruses. Moreover, the transportation of these samples via delivery vehicles traveling between testing centers and community sites elevates the risk of contact with infected individuals, potentially turning vehicles into carriers of the virus. This increased exposure and potential for the spread of virus through sample movement highlights the importance of implementing the necessary measures to minimize the risk of infection transmission during transport. Addressing this challenge requires the development of an innovative sample collection method that prioritizes effectiveness while minimizing virus transmission risk. Ideally, this new method should be rapid and automated, facilitating early epidemic prevention actions and mitigating gatherings and cross-infection during collection and transportation. Thus, it is meaningful to propose an infection sample collection system. The development of technologies such as IoT and blockchain provide feasible solutions for this work.

In recent years, the technology behind drones has undergone rapid development, resulting in significant improvements in endurance and payload capacity [5]. As a result, drones offer a new solution for the collection of infectious disease samples. Many companies have attempted to use drone delivery technology for sample transport. In Hangzhou, China, drones are already being utilized for the transportation of nucleic acid samples (Figure 1). During the epidemic period, up to 100,000 samples can be delivered daily. The employed drones can support a payload of 10 kg, a flight time of at least 20 min, and a range of 18 km. Compared to traditional vehicle transportation, using drones for sample delivery has several advantages: firstly, replacing traditional modes with unmanned systems saves medical resources; secondly, transporting samples by drone avoids the temporal and spatial interaction between sample transfer and individuals, thereby reducing the risk of epidemic spread during transportation to the greatest extent possible; thirdly, drones can avoid the impact of traffic control and terrain factors on sample transport, enabling even more timely delivery; and finally, drones can be equipped with sensors, cameras and other electronic devices to ensure sample safety by recording the transportation process [6].

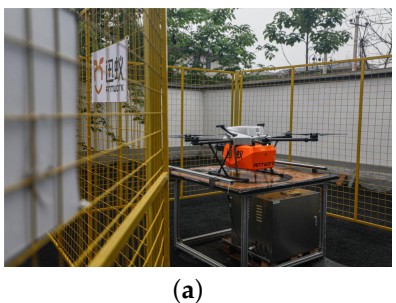 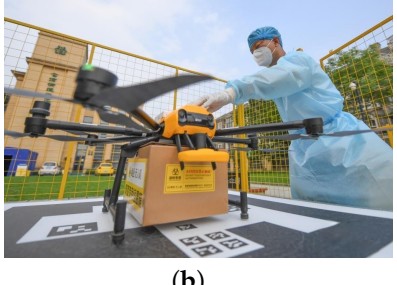

(**a**)                    (**b**)

**Figure 1.** Drones are utilized for the transportation of nucleic acid samples in Hangzhou, China. (**a**) Drones arrive at designated locations to load samples. (**b**) Medical personnel load samples onto the drone.

However, existing drone-assisted sample collection solutions still have the potential to improve the timeliness and safety of sample collection [7]. Additionally, existing solutions have significant shortcomings in terms of epidemic information security and the risk of epidemic spread. Drones are only used for point-to-point sample transportation between sampling points and testing centers, rather than participating in the entire collection task. Participants still need to personally queue up at sampling points, which increases the risk of personnel gathering and cross-infection. The traceability of infectious disease samples is also difficult to realize, resulting in medical institutions being unable to take effective measures to control the spread of epidemics. Moreover, public health information that lacks management and protection is at risk of being tampered with. Essentially, in existing drone-assisted sample collection solutions, drones are only replacing vehicles in the task mode, without any fundamental difference. In addition, regarding sample collection, remote areas

have always been a challenge due to sparse populations and inadequate infrastructure. For these scenarios, both existing drone-assisted sample collection solutions and traditional shuttle-based sample collection solutions make it difficult to achieve the safe and rapid collection and delivery of samples.

To overcome the shortcomings discussed and improve the safety and timeliness of sample collection, we integrate emerging technologies such as IoT, drones, and blockchain to propose the Blockchain-enabled Infection Sample Collection system (BISC). The blockchain's chain structure improves information security and makes infection samples traceable. We also propose a two-echelon drone-assisted mechanism that utilizes two types of drones for transporting samples to testing centers: the agile and fast collector drone for collecting infection samples at user points, and the deliverer drone for long-distance sample delivery. The brief workflow of the system is as follows: The collector drone will transfer the samples to the designated transit points, where they will be handed over to the deliverer drone. The deliverer drones receive samples at transit points and transport them to the nearest testing center. All samples are stored and tested at the testing center. This solution realizes contact-less sample collection so that users complete sampling without the extra risk of infection. The data generated during the operation of the system and the test results are stored in the blockchain to ensure data security. The two types of drones can effectively reduce the risk of cross-infection during sample collections due to different activity areas, thus improving the biological safety and reliability of sample collection. In addition, the collaboration between collector drones and deliverer drones can significantly improve the effectiveness of sample transmission in remote areas. Our main contributions can be summarized as follows:

- We propose a blockchain-enabled infection sample collection system that is built with an infrastructure layer, a blockchain layer, and an application layer. The information security of infection samples is ensured using the hash function, chain structure, timestamp, and consensus mechanism. The routing algorithm for sample collection is integrated into the smart contract to improve the reliability and automaticity of BISC.
- We propose a model of a Two-Echelon Heterogeneous Drone Routing Problem with Transit point Synchronization (2E-HDRP-TS). The model integrates fixed transit points, synchronized handover operations, and heterogeneous drones with limited energy and capacity. It is specifically designed for collecting infection samples from all user points with two types of drones. It improves the time efficiency of samples, especially in remote areas, and reduces the probability of cross-infection during sample collection. To measure sample safety, we introduce the exposure index.
- We present a multi-objective Adaptive Large Neighborhood Search algorithm for Routing of Drones (ALNS-RD). By incorporating flight distance-based and exposure index-based operators, this algorithm can better plan the drone's route and improve the total collection time for infection samples and the exposure index. Based on the dominance relationship, we refine the criteria for multiple objectives and adaptive mechanisms. To increase diversity in the Pareto set, we determine the search direction based on the crowding distances.
- We conducted a series of experiments based on practical cases. First, we optimized and adjusted the key parameters used in ALNS-RD. Then, we validated the effectiveness and superiority of using the flight distance operator and exposure index operator to improve the quality of the Pareto set through experiments. Finally, we compared the proposed ALNS-RD algorithm with existing ones (i.e., NSGA-II, MOLNS). The experimental results demonstrated that ALNS-RD performs much better when solving the 2E-HDRP-TS for infection sample collection.

The rest of this article is organized as follows. Section 2 discusses related work, Section 3 describes the architecture of BISC, Section 4 describes the two-echelon drone-assisted mechanism, Section 5 specifically addresses the solution approach, Section 6 provides the experimental results, and Section 7 concludes the article.

## 2. Related Work

With the continuous performance improvement of drones, they have demonstrated significant advantages in efficiency and flexibility. Research on the application of drones in logistics delivery is also increasing. We first review the literature on drone delivery systems and then review the literature on 2E-VRP and corresponding solution methods.

### 2.1. Drone-Assisted Delivery System

Due to drones' enormous potential, extensive research has been conducted on drone delivery. Andreato et al. [8] demonstrated the feasibility of drone delivery from a cost perspective and highlighted the challenges that must be overcome to make it practical. Sundar et al. [9] proposed a routing problem that considered the fuel constraints of a single drone while minimizing total fuel and enabling multiple refueling stations. Song et al. [10] proposed a scheduling model for a drone-delivery logistics system, which allows drones to share multiple charging stations due to the constraints of the load capacity and flight time. Dorling et al. [11] proposed a Drone Delivery Problem (DDP), implementing a mixed-integer linear program that considered battery weight, payload weight, and drone reusability. Tian et al. [12] proposed the Solar-Powered UAV Delivery System (SPU), which eliminates the need for charging facilities. When the SPUs run out of power, they recharge themselves at a designated landing site instead of at charging stations. Other studies have used a combination of drones and ground vehicles for delivery. Murray et al. [13] first introduced drones into the study of the Traveling Salesman Problem (TSP), calling it the Flying Sidekick Traveling Salesman Problem (FSTSP), in which drones assist vehicles in completing package deliveries and specifying that drones only serve one customer in a single journey. Ferrandez et al. [14] established a mathematical model to expand the truck–drone delivery system by optimizing the number and location of drone launch sites. Sajid et al. [15] proposed mixed-integer linear programming models for UAV-routing and UAV-route scheduling problems, taking into account the effect of incidental processes and the varying payload on travel time. Aggarwal et al. [16] analyzed different types of UAV path planning techniques and discussed issues related to UAV network communications, providing a comprehensive research perspective. Laghari et al. [17] investigated the design of UAV systems, especially communication systems, and proposed sensor-based data processing methods as well as applications. Deng et al. [18] proposed a new solution for vehicle-assisted UAV delivery, allowing UAVs to serve multiple customers in a single take-off while considering energy consumption. She et al. [19] proposed self-organized UAV traffic flow in low-altitude 3D airspace, formulating the user equilibrium condition using partial differential equations and proposing a scheme for numerical solution and system performance computation.

In the field of medical delivery, many studies have shown that drones can improve the efficiency and cost-effectiveness of blood delivery, laboratory testing, and medication delivery [20,21]. Hii et al. [22] pointed out that the process of drone medical delivery can potentially affect the quality of medical supplies. In their study, they indicated that medical supplies such as insulin are very sensitive to the environment. Therefore, the potential impact of temperature and vibration on the effectiveness of medical supplies should be fully considered in drone-assisted medical delivery. Scott et al. [23] designed a new model for drone medical delivery systems. The model combines traditional land transportation with drone delivery and uses budget as a constraint to provide location decisions for drones. In addition, as the variety of IoT devices increases and technology advances, the integration of medical systems and IoT becomes a feasible solution. Zeadally et al. [24] presented the devices and technologies that need to be incorporated into the IoT ecosystem for medical systems. They emphasize that providing reliable connectivity for medical devices and sensors through the IoT ecosystem is the key to unlocking the digital medical system. Paganelli et al. [3] proposed a detailed description of an IoT-based conceptual architecture for a COVID-19 patient monitoring system. The architecture also provides modules for analyzing health data to provide patients with feedback and insights. Otoom et al. [25]

proposed a solution for the detection and monitoring of COVID-19 cases, which collects real-time data from wearable sensors and uses machine learning to predict suspected cases. We summarize the comparison between our work and the existing literature in Table 1.

**Table 1.** Comparison between our work and the existing literature.

| Reference | Complete Physical Model | Route Scheduling | Medical Item | Blockchain |
|:---:|:---:|:---:|:---:|:---:|
| [8,17] | ✓ | | | |
| [9–11,15,16] | ✓ | ✓ | | |
| [12–14] | | ✓ | | |
| [20–22] | ✓ | | ✓ | |
| [23] | ✓ | ✓ | ✓ | |
| [3,24,25] | | | ✓ | |
| Our work | ✓ | ✓ | ✓ | ✓ |

Currently, drone-assisted delivery is a hot topic in the logistics field. However, research on drone-assisted delivery of medical items, especially with regard to the delivery of infection samples, is still relatively rare. Unlike ordinary delivery objects, medical items often have characteristics such as being fragile and susceptible to contamination. Compared with traditional delivery processes, medical delivery not only considers costs and time but also the safety and effectiveness of medical items. In particular, for drone-assisted delivery of infection samples, we need unique transportation strategies to minimize the exposure risks of the samples. In addition, we also need to consider the drone's energy consumption and capacity to ensure that it can successfully complete the delivery task. In addition, as far as we know, there are no papers on systems combining blockchain and drone collection of infection samples. The introduction of blockchain is beneficial for improving the traceability and information security of infection samples.

### 2.2. 2E-VRP

The two-echelon drone-assisted mechanism described in this paper can be characterized as a two-echelon heterogeneous drone routing problem with transit point synchronization, which is a variant of the two-echelon vehicle routing problem (2E-VRP). The 2E-VRP has been widely studied in academia over the past decade, and the classic model is as follows: there is a central station and a fixed number of intermediate sites with operational capacity restrictions on a two-level logistics network; a specific number of vehicles are used on each level of the network; the demand of all customers is fixed and known, and cannot be divided. The 2E-VRP has attracted widespread attention in the past decade, and the methods for solving this problem mainly include exact algorithms and heuristic algorithms. González-Feliu et al. [26] were the first to propose the model of 2E-VRP, and solved an instance with 32 customers and satellites using the branch-and-cut algorithm. Perboli et al. [27] introduced some new optimality cut classes. They improved the algorithm by adding valid inequalities based on the network flow formulation and the connectivity of the transportation system graph. Jepsen et al. [28] proposed a modified branch-and-cut algorithm to address the problem of an incorrect upper bound when the number of transfer stations exceeds two in Perboli et al.'s model [27]. Baldacci et al. [29] proposed a new mathematical model in which two binary variables were used to decide whether a feasible route was adopted. They proposed an exact algorithm that decomposes the 2E-VRP into a set of VRPs with multiple depots, further enhancing the scalability and accuracy of the algorithm for solving instances. If the problem size expands further, exact algorithms cannot be efficiently solved. Therefore, in recent years, researchers have turned to using heuristic algorithms and metaheuristic algorithms to solve the 2E-VRP. Hemmelmayr et al. [30] proposed the Adaptive Large Neighborhood Search (ALNS) heuristic algorithm, which is specifically designed for the structure of the 2E-VRP. Breunig et al. [31] proposed a hybrid metaheuristic that combines enumeration local search with destruction and repair heuristics. Compared with other algorithms, the

ALNS algorithm has a greater advantage in solving problems due to its extensive heuristic operators providing directions for neighborhood search.

In recent years, many improved models have emerged based on the classic model of 2E-VRP. Regarding the two-echelon vehicle routing problem, Crainic et al. [32] first proposed a 2E-VRP model with a synchronization concept, which integrated multiple depots, multiple tours, heterogeneity, and time windows. Perboli et al. [33] proposed a two-echelon vehicle path planning problem with satellite synchronization. Granger et al. [34] proposed the 2E-VRP problem with time windows, satellite synchronization constraints, and multi-trip based on urban scenarios. Li et al. [35] proposed the two-echelon vehicle path planning problem with satellite dual synchronization and solved it with an improved ALNS algorithm. Belgin et al. [36] proposed a two-echelon vehicle routing problem with simultaneous pickup and delivery (2E-VRPSPD) and developed a hybrid heuristic algorithm based on variable neighborhood descent and local search. Chen et al. [37] proposed a new two-echelon delivery system based on hybrid machine learning and heuristic optimization. The method pre-distributes predicted demands from depots to regional facilities to improve the solution quality. Yu et al. [38] proposed the two-echelon van-based robot routing problem with hybrid pickup and delivery, including five new one-to-one pickup and delivery modes. Paredes-Belmar et al. [39] proposed a milk transportation problem with vehicle routing and the location of milk collection centers. They addressed the location of milk collection centers to reduce costs. Xue et al. [40] proposed a two-echelon dynamic vehicle routing problem with proactive satellite stations, which converts customers with available idle storage to satellite stations and optimizes the transportation cost and storage cost.

The above research mainly focuses on the delivery problem based on vehicle transportation and has made remarkable progress in problem modeling and algorithm design. However, these models and algorithms are not applicable to the infection sample collection scenario: (1) existing research mainly takes delivery time and cost as the main optimization objectives. However, in the sample collection scenario, the exposure level of the sample during the delivery process is a crucial factor that must be considered in addition to the above two factors. Therefore, we should minimize the exposure risk of the samples by minimizing their residence time in the handover link; (2) existing research only addresses the problem of vehicle routes without considering the relationship between effective load and energy consumption, but in drones-assisted sample collection, we must consider the impact of energy consumption and payload on the drone routes. Therefore, in this paper, we propose a 2E-HDRP-TS model. In this model, we consider factors such as the degree of cross-infection and the collection time of samples during the delivery process. To measure the safety of the samples, we propose the concept of exposure index and use it as one of the optimization objectives. Also, we focus on minimizing the collection time to improve the delivery efficiency. Our goal is to find the optimal delivery route while satisfying various constraints.

## 3. Architecture of BISC

In this section, we describe the architecture of BISC in detail.

### 3.1. Overview of System

Infection samples are currently an essential and valued material for interrupting the early spread of the virus. BISC typically conducts regular sample collection from high-risk communities at the initial stage of an outbreak. This critical link requires a high level of safety and timeliness. In terms of biological safety, all infection samples collected from users should be considered potentially pathogenic, so it is necessary to take precautions against the cross-infection from samples. BISC uses heterogeneous drones and a two-echelon mechanism is used collecting infection samples. Drones flying to user points in high-risk areas are more susceptible to contamination. Therefore, they are only allowed to fly to transit points to unload samples and do not have access to the testing center. The

location of the testing center in a low-risk area far away from residents is conducive to avoiding the risk of cross-infection of infection samples and ensures accurate sample test results. In the BISC system, biological safety and information security are inseparable, yet traditional drone-assisted collection systems have serious information risks. The infection sample data containing sensitive personal information lack encryption and protection. It is susceptible to attacks that could result in privacy breaches. In addition, as important public health information, the infection sample data require traceability and verifiability. Infection samples lack integrity and authenticity if the source and modification of information cannot be recorded. The biological safety cannot be guaranteed either. Blockchain, as a distributed, decentralized, tamper-proof, and traceable technology, can provide an effective solution for information security [41]. Therefore, we propose an infection sample collection system with the integration of blockchain. In such a blockchain-enabled system, distributed nodes are able to exchange the required data without the need for a third-party intermediary by reaching a consensus. Sample numbers, container selection, event documentation, and transportation environments all have significant impacts on the biological safety of samples. Blockchain uses encryption algorithms and digital signatures to ensure the security and privacy of the above information. The hash value and timestamp are used to record the source of and changes in information [42].

A typical application scenario of BISC is shown in Figure 2. We aim to accomplish the collection, transportation, and testing of infection samples in remote communities with sparse populations. The public health information is also guaranteed to be traceable and tamper-proof. Heterogeneous drones are used for infection sample collection and delivery. Collector drones fly precisely to user points, completing the "last meter" of the collection task. This mechanism essentially avoids the possibility of spreading the virus between residents. The other type of drones are deliverer drones, which are mainly responsible for transporting samples to the testing center after collector drones have completed a certain amount of collection. Finally, all sample data and test results are stored in the blockchain to ensure information security. Reliable sample information not only benefits the tracing and prevention of infectious diseases, but also provides verifiable data support for public health management. Through blockchain technology, we can ensure transparent records of sample origins and modifications, thereby maintaining the integrity and authenticity of infection samples. This measure is not only important for biological safety, but also helps establish the credibility and traceability of public health information, providing crucial support for responding to disease outbreaks.

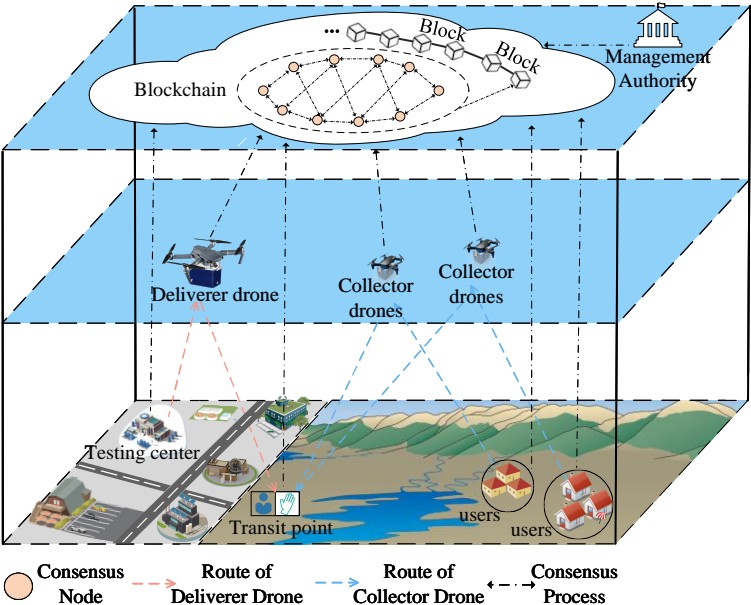

**Figure 2.** Overview of BISC.

### 3.2. Composition of BISC

The composition of BISC consists of four participating entities, including management authority, medical institutions, individual users, and transportation agencies, as shown in Figure 3.

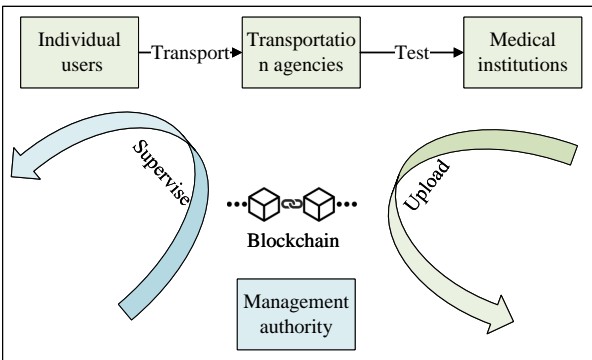

**Figure 3.** The participants of BISC.

(1) **Management authority**: The management authority is responsible for the identity registration of all entities, as well as the distribution of keys, and takes a leading role in BISC. In addition, throughout the entire collection task, it reads the information on infection samples in order to facilitate timely decision-making for preventing virus spread.

(2) **Medical institution**: The medical institutions include hospitals, testing centers, and laboratories that have the capability to test samples. In BISC, the medical institutions are responsible for the testing of infection samples and the uploading of the results. Naturally, the medical institutions are also responsible for epidemiological analysis of the infectious disease infection and for providing mitigation strategies.

(3) **Transportation agency**: The transportation agency, which consists mainly of drones and transit points, undertakes the task of collecting and transporting samples throughout the entire collection task. Both the collector drone and the deliverer drone are identified after reaching the transit point to ensure that the infection samples are not illegally accessed. In addition, transportation data, including the flight routes of the drones and handover operations, are uploaded into the blockchain by the drones.

(4) **Individual user**: The individual users accomplish the binding of the personal sample to the sampling container via the Internet. Also, before delivering the sample, the individual user needs to verify the identity of the drone. The data generated from the above delivery process are uploaded into the blockchain along with basic personal information. Naturally, for privacy protection, individual users can only access their own sample test results.

The workflow of BISC is shown in Figure 4. First, the management authority identifies and registers the other entities of the system and assigns the keys. The collector drones collect infection samples according to a flight route set out by the system. When a collector drone arrives at a user point, it must be identified before it takes a sample from that point. Then, it transports several samples to a transit point, waits for the deliverer drone to arrive, and completes the sample handover after another identity verification. Deliverer drones transport a large number of samples to a medical institution. All the above transportation information is packaged and uploaded to the blockchain. The medical institution will also upload the corresponding test results of the sample. In this way, individual users can check their health status through the key. The management authority and the medical institution, on the other hand, establish precautionary measures as soon as possible based on the infection status of the residents in the region.

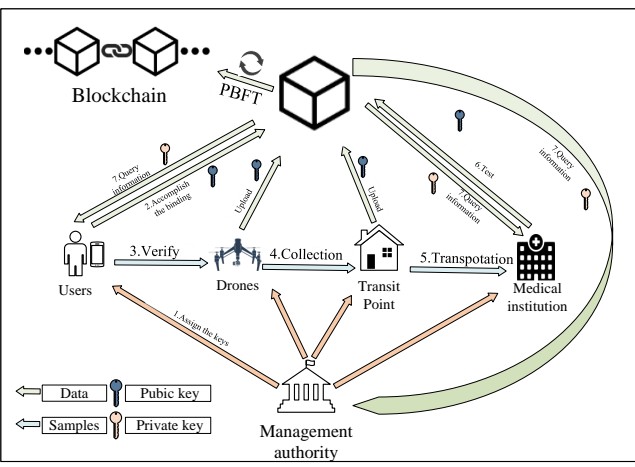

**Figure 4.** System workflow.

### 3.3. Layers of BISC

The BISC connects the management authority, medical institutions, individual users, and even transportation agencies into a network. To ensure information security, we adopt an alliance chain in BISC, which allows several nodes within different organizations to join together [43]. We propose a three-layer architecture consisting of an infrastructure layer, a blockchain layer, and an application layer, as shown in Figure 5.

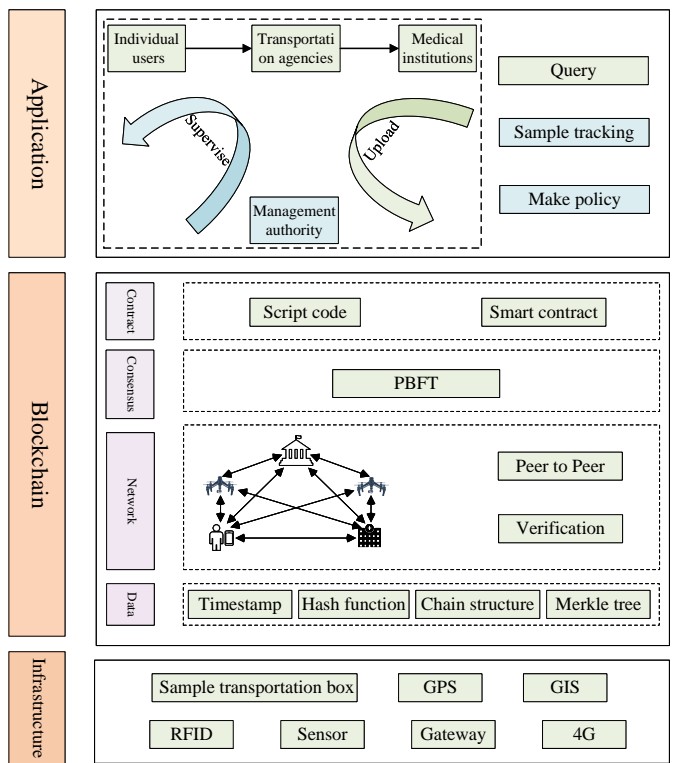

**Figure 5.** The architecture of BISC.

(1) The infrastructure layer is the underlying layer of the system architecture. The infrastructure refers to the basic physical facilities and equipment used to collect information, including the Global Positioning System (GPS), cameras, sample transportation boxes, Radio Frequency Identification (RFID), and a variety of sensors. The system may record route data for the collection process (including information on drone paths and handover operations) via GPS and cameras. During the transportation of infection samples, the environment of the sample transportation box affects the

reliability of sample testing. Therefore, sensors mounted inside the box record data on the environment (temperature and vibration during collection) and monitor deviations in the collection procedure from standard operating procedures (for example, a longer collection time or a temperature other than the one recommended). RFID improves the efficiency of identity verification between drones and user points. The infrastructure layer mainly provides hardware support for the collection of infection samples with accurate recording of route data and environmental data throughout the collection process.

(2) The middle layer of the system is dedicated to the blockchain, in which all system data are organized into a chronological sequence of interconnected blocks. This layer incorporates a peer-to-peer network and consensus mechanism to uphold data consistency across nodes. Moreover, smart contracts are implemented to enhance information management and optimize infection sample collection routes. Within the blockchain layer, distinct components such as the data layer, network layer, consensus layer, and contract layer interact seamlessly to ensure the robust operation of the system. Each layer has specific capabilities, as follows:

**Data layer**: The data stored in the blockchain mainly include the user identity, samples information, route of drones, and testing results. When any node needs to update the data, it sends a data update request to BISC to verify the transaction by consensus. Only the data that pass the consensus mechanism can be packaged into a data block, which is linked to the original data block to form a chain structure. During this process, technologies such as timestamps, hash functions, and Merkel trees are applied to keep the information secure and prevent tampering [43].

**Network layer**: The network layer of the blockchain is a peer-to-peer (P2P) structure which is used for communication and authentication. The use of P2P technology ensures that all participating entities in the system can take part in the transaction. When participating entities need to synchronize data, they broadcast messages to other nodes in the network for consensus.

**Consensus Layer**: In BISC, how nodes reach a consensus is key. In this paper, we use the Practical Byzantine Fault Tolerance (PBFT) consensus algorithm. Its main principle is to take the management authority of BISC as the master node, while all other nodes are child nodes. For every consensus computation, all the nodes in the system communicate with each other and reach a consensus based on the principle of majority rule. We chose the PBFT algorithm to be applied to the infection sample collection system for the following reasons: Unlike the Proof Of Work (POW) algorithm, the PBFT algorithm does not consume a large amount of computing resources, and its efficiency in dealing with transactions is improved. Moreover, for each consensus computation, the PBFT algorithm allows for, at most, $(N - 1/3)$ invalid or malicious nodes. That means, among all the nodes in the system, it must be guaranteed that there are at least $(2N + 1)/3$ normal nodes. Most of the participating subjects in BISC are hospitals, testing centers, or government administrations. They are much less likely to execute malicious behaviors than other nodes due to strict network security measures. The feature provides a relatively safe and stable environment base for the normal operation of the PBFT algorithm.

**Contract layer**: The contract is the core of the whole system, consisting of script code and smart contracts. The smart contract is a special program that needs to formulate the trigger conditions in advance. When the executions are triggered, the corresponding contract codes are executed automatically, and the outside world does not need to and cannot intervene manually [44]. This increases the trustworthiness of the system. In BISC, the smart contract is designed to improve sample information management and plan a safe and efficient sample collection path.

(3) The application layer serves as a platform that incorporates the blockchain system to facilitate the exchange of information among diverse participating entities, including management authorities, medical institutions, individual users, and transportation

agencies. For example, individual users can query their health information through the application layer. The management authority can also find out the scale of infection in the region and issue public health measures.

## 4. The Two-Echelon Drone-Assisted Mechanism

The BISC uses blockchain to address the issues of traceability and information security. However, in addition to security, it is also necessary to ensure the effectiveness of the sample collection and the reliability of the handover process. Routing is key to improving the effectiveness and reliability during the collection. Therefore, we present a two-echelon drone-assisted mechanism, which is characterized as a 2E-HDRP-TS. In this section, we first present the 2E-HDRP-TS, followed by a mathematical model of the problem. For ease of understanding, Table 2 summarizes the notation used in the 2E-HDRP-TS model.

**Table 2.** Model notations.

| Notation | Notation Description |
| --- | --- |
| $V_u$ | The set of all user points |
| $K_1$ | The set of deliverer drones |
| $K_2$ | The set of collector drones |
| $V_s$ | The set of transit points |
| $G(V, A)$ | The directed graph for all drones |
| $G_1(V_1, A_1)$ | The directed graph for first-level deliverer drone |
| $G_2(V_2, A_2)$ | The directed graph for second-level collector drone |
| $q$ | The sample quality of user points |
| $TC$ | The testing center |
| $D_{i,j}$ | The distance between two users |
| $t_{i,j}^D$ | The flight time of deliverer drone between two users |
| $t_{i,j}^C$ | The flight time of collector drone between two users |
| $L_i^D$ | The load of the deliverer drone leaving point $i$ |
| $L_i^C$ | The load of the collector drone leaving point $i$ |
| $H_1$ | The maximum capacity of the deliverer drone |
| $H_2$ | The maximum capacity of the collector drone |
| $E_1$ | The maximum energy of the deliverer drone |
| $E_2$ | The maximum energy of the collector drone |
| $w_1$ | The weight of the deliverer drone |
| $w_2$ | The weight of the collector drone |
| $P_1$ | The maximum power of the deliverer drone |
| $P_2$ | The maximum power of the collector drone |
| $F_{i,j}^D$ | The energy consumption of deliverer drone between two users |
| $F_{i,j}^C$ | The energy consumption of collector drone between two users |
| $v_{i,j}^D$ | The flight velocity of deliverer drone between two users |
| $v_{i,j}^C$ | The flight velocity of collector drone between two users |
| $T$ | The total time to complete sample collection |
| $Q$ | The exposure index of the entire solution |
| $x_{i,j}^k$ | The decision variables between $i$ and $j$ |

### 4.1. Model Descriptions

4.1.1. 2E-HDRP-TS Model

We target remote communities in which BISC is applied. Within this scenario, users are sparsely distributed and are located far from the infection sample testing center. To reduce the time spent on sample collection and eliminate the biological safety issues associated with traditional sampling methods, two-echelon heterogeneous drones are used for the collection task. We classify all user points and infection samples as potentially pathogenic, thus delineating high-risk and low-risk areas. Drones traveling to user points are susceptible to virus contamination in high-risk areas and are therefore restricted to unloading samples at designated transit points rather than entering the testing center.

Situating the testing center in a low-risk area far from users effectively mitigates the risk of cross-contamination of infection samples and ensures the accuracy of sample test results. For the above scenario, we propose the 2E-HDRP-TS, in which the transit point is one or more designated locations used for sample handover between drones. As shown in Figure 6, the 2E-HDRP-TS includes two types of drones, each of which may create multiple delivery routes. One type of drones, called collector drones, is fast and flexible. They are responsible for collecting samples from various user points and transporting them to the transit points. The other type of drones, called deliverer drones, have a larger weight and stronger load capacity. They shuttle between the testing center and the transit points to transport packaged infection samples to the testing center. Here, we assume that all infection samples are returned to the testing center and tested there.

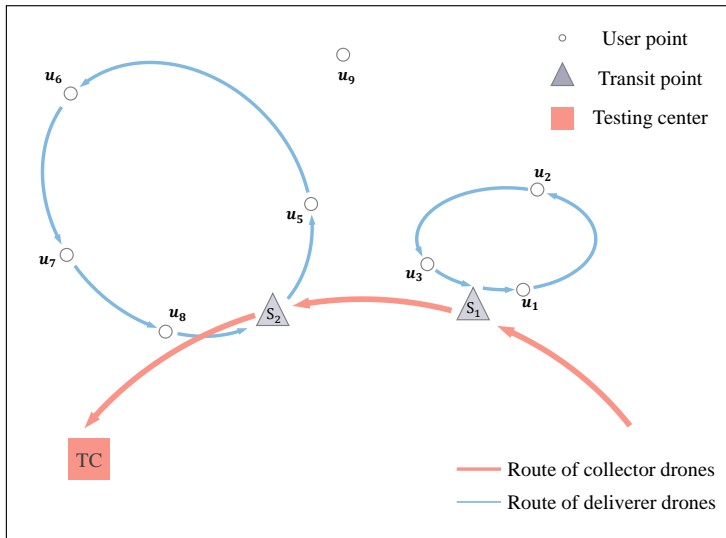

**Figure 6.** Two-echelon collection process.

The purpose of this study is to plan the routes of multiple drones to safely and quickly deliver infection samples to a testing center. A directed graph $G(V, A)$ that contains all the flight routes of drones is given, which includes the first-level delivery graph $G_1(V_1, A_1)$ and the second-level collection graph $G_2(V_2, A_2)$. The deliverer drones travel between the testing center (TC) and transit points, and between the transit points. We define $V_1 = \{CT\} \bigcup \{V_s\}$ and $A_1 = \{(CT, i) | i \in V_s\} \bigcup \{(i, j) | i, j \in V_s\} \bigcup \{(i, CT) | i \in V_s\}$. The collector drone sets off from a transit point, flies to the users in sequence to collect samples, and finally flies to the transit point for synchronous handover. In the same way, we define $V_2 = \{V_u\} \bigcup \{V_s\}$ and $A_2 = \{(i, j) | i \in V_s, j \in V_u\} \bigcup \{(i, j) | i, j \in V_u\} \bigcup \{(i, j) | i \in V_u, j \in V_s\}$.

4.1.2. Energy Consumption Model of Drones

In the 2E-HDRP-TS model, the weight of the samples collected by drones can approach or even exceed the self-weight and cannot be ignored. Therefore, the energy consumption of the drone is determined by both its own weight and the payload. As the drone collects samples one by one, the increased payload also increases the energy consumption rate. Finally, the collection route must be completed within the energy limit. This is to prevent the drone from being unable to complete the scheduled route due to insufficient energy supply, which could result in unreliable or even destroyed infection samples. The payload changes for each point the drone passes through, either for sample collection or sample loading and unloading tasks. Thus, the energy consumption rate between two nodes shows

a stepped change. Taking the collector drone as an example, the power of the battery when flying from node $i$ to node $j$ is as follows [8]:

$$p_{i,j}^C = \frac{(w_2 + L_i^C)v_{i,j}^C}{370\eta\gamma} + e, \tag{1}$$

where the self-weight of the collector drone is $w_2$, the payload is $L_i^C$, the speed is $v_{i,j}^C$, $\eta$ is the power transfer efficiency for motor and propeller, $\gamma$ is the lift-to-drag ratio, and $e$ is the energy consumption of other electronic devices on the drone.

In addition, to enhance the delivery efficiency of the drones, we assume that the drones maintain the maximum power during flight, $p_{i,j}^C = P_2$ [45]. Based on this, the formula for the flight speed and travel time with the payload $L_i^C$ can be calculated by

$$v_{i,j}^C = \frac{370\eta\gamma(P_2 - e)}{w_2 + L_i^C}, \tag{2}$$

$$t_{i,j}^C = \frac{D_{i,j}}{v_{i,j}^C} = \frac{D_{i,j}(w_2 + L_i^C)}{370\eta\gamma(P_2 - e)}. \tag{3}$$

The parameters in Equations (2) and (3) can be determined based on the specific drone model in the experiment. Figure 7 shows the variation of the collector's velocity on the way to complete the sample collection task.

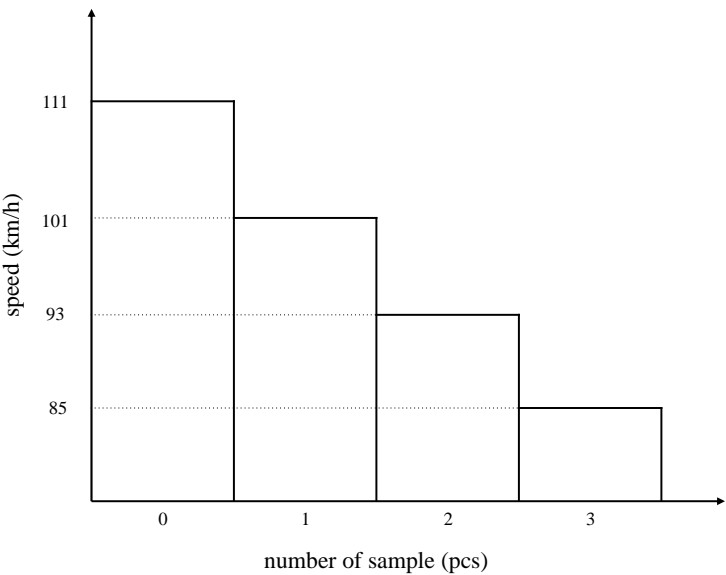

**Figure 7.** Relationship between velocity and payload.

The energy consumption of a drone also varies with its payload, which is defined as

$$F_{i,j}^C = P_2 \times t_{i,j}^C = P_2 \frac{D_{i,j}(w_2 + L_i^C)}{370\eta\gamma(P_2 - e)}, \tag{4}$$

where $F_{i,j}^C$ represents the energy consumption of the collector drone flying from use point $i$ to use point $j$ with payload $L_i^C$. Obviously, the energy consumption model of the deliverer drones follows the same principle. When all the collector and deliverer routes are known, we can calculate the following information: (1) the total energy consumption of each route, which can be used to determine whether it exceeds the energy limit; (2) the arrival time at the transit point, which can be used to determine the synchronous performance during handover operation; and (3) the total time taken to complete sample collection. To simplify

the model, we ignore the speed changes caused by the takeoff and landing of the drone. In addition, we do not conduct research on factors such as weather conditions.

4.1.3. Sample Exposure Index

In this section, we first introduce the handover operation and describe the handover process in detail, and then we develop a performance index called the exposure index to assess the cross-infection risk of a single synchronous handover operation.

Synchronous handover: In the 2E-HDRP-TS, we use two types of drones for sample collection. When the collector drone and deliverer drones arrive at the transit point, they perform a handover operation. The handover operation is defined as medical personnel unloading samples from the collector and loading them onto the deliverer. At this time, the security of the samples is threatened, and samples are easily contaminated or even destroyed. So, the handover process becomes the key to preventing sample cross-infection. We should minimize the waiting time of drones carrying samples at the transit point and the time spent on handover operations. Therefore, we require the two drones to be synchronized as much as possible. In addition to waiting for the other drone, we should also consider the following time costs: (1) brief disinfection of the drone load box, preparation for handover; (2) unloading and loading operations; (3) preparation for takeoff.

As shown in Figure 8, we use the example of a transfer operation where the collector drone arrives first. $[t_b, t_c]$: The drones identify each other and prepare for the handover. $[t_c, t_d]$: The unloading and loading time of the two drones. During a single handover operation, two drones must be present at the same time, and the collector's sample is unloaded and loaded onto the deliverer. $[t_d, t_e]$: The deliverer drone prepares for takeoff. However, we assume that the above time can be ignored and will not affect the solution's performance. This is because the identity verification and unloading operations of the drones in the proposed model are very simple and take little time. We only consider the arrival times of the two drones, $t_a$ and $t_b$, and both drones leave at $t_b$. Based on this, we can describe the exposure index.

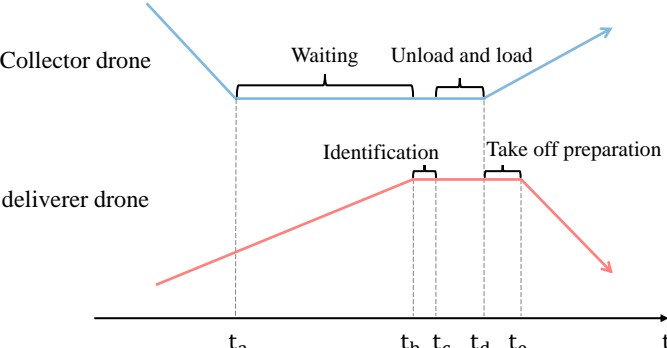

**Figure 8.** Time chart for a handover operation.

Exposure index: If the time difference between the two drones (the collector drone and the deliverer drone) arriving at the designated transit point is too large, the infection samples carried by these two drones will inevitably be at high risk of exposure. Therefore, in order to measure the synchronous performance and sample safety of a single handover operation, we propose an exposure index $Q_i$. The exposure index $Q_i$ is divided into two parts, namely $Q_i^{RE}$, which measures the degree of sample retention, and $Q_i^{SY}$, which measures the degree of synchronization. $Q_i^{RE}$ is obtained by multiplying the quality of the retained sample by the retention time. $Q_i^{SY}$ is obtained by multiplying the time difference by the average payload factor. The higher the exposure index, the higher the danger and unreliability of the sample. The exposure index $Q_i$ is defined as follows:

$$Q_i^{RE} = L_i^D * \left( T_i^{ON} - T_i^{DA} \right) + L_i^C * \left( T_i^{ON} - T_i^{CA} \right), \tag{5}$$

$$Q_i^{SY} = k_{SY} * \left| T_i^{DA} - T_i^{CA} \right|, \tag{6}$$

$$Q_i = Q_i^{RE} + Q_i^{SY}, \tag{7}$$

where i represents the *i*-th handover operation, $T_i^{CA}$ is the time at which the collector arrives, and $T_i^{DA}$ is the time at which the deliverer arrives. $T_i^{ON}$ is the time at which the deliverer completes all handover operations at that point, and $k_{SY}$ is the average payload factor, measured in kg. During a handover operation, handover can only occur when both drones have arrived, and after unloading and loading their respective samples, they will depart. In addition, if a different collector visits a handover point corresponding to multiple different operations, the exposure index needs to be calculated separately. The exposure index of the entire plan is the average of the exposure indexes of all transit operations.

$$Q = \frac{\sum_i^B Q_i}{B}, \tag{8}$$

where *B* represents the total number of handover operations.

*4.2. Problem Formulation*

In the 2E-HDRP-TS, the positions and number of user points are known, and the quality of samples generated by all user points is equal. Several collector drones are needed to transport all users' samples to the transit points, which requires collectors to collect multiple users in a single flight. The transit point is one or more predetermined locations used for sample transfer between drones. Subsequently, the deliverer drones transport the samples to the testing center. The speed of the drones constantly changes with the load. There is no storage capacity at the transit point, so both drones should try to arrive at the same time during the handover operation to reduce the risk of sample cross-infection. Ultimately, our goal is to find the optimal flight routes for drones to minimize the collection time and exposure index as much as possible. The total completion time is the time at which all deliverers arrive at the testing center, which is defined as

$$T = \max(T_i, \forall i \in K_1), \tag{9}$$

where $K_1$ represents the set of deliverer drones and $T_i$ represents the time at which the *i*-th deliverer drone returns to the testing center.

To simplify the problem, we make the following assumptions: (1) we assume that all collector drones are homogeneous, and the deliverer drones are the same; (2) the start-up, acceleration, and stopping of the drones can be ignored; (3) the power of the drones is constant; (4) during the transit process, the loading and unloading of the drone is very simple, and we will ignore this part of the time. In the 2E-HDRP-TS, we aim to optimize the collection duration and exposure index. The problem is a typical multi-objective optimization problem, defined as

$$\min(f_1, f_2), \tag{10}$$

$$st. \sum_{(tc,j)\in A_1} x_{tc,j}^k = \sum_{(tc,j)\in A_1} x_{j,tc}^k = 1, \forall k \in K_1, \forall j \in V_s, \tag{11}$$

$$\sum_{i\in V_s, j\in V_u} x_{i,j}^k = \sum_{i\in V_s, j\in V_u} x_{j,i}^k = 1, \forall k \in K_2, \tag{12}$$

$$\sum_{(i,j)\in A} x_{i,j}^k = \sum_{(j,i)\in A} x_{j,i}^k, \forall k \in K, \ \forall i \in V, \ \forall j \in V, \tag{13}$$

$$\sum_{K\in K_2} \sum_{(i,j)\in A_2} x_{i,j}^k = 1 \ , \ \forall i \in V_2, \forall j \in V_u, \tag{14}$$

$$0 \leq L_i^D \leq H_1, \forall i \in V_1, \tag{15}$$

$$0 \leq L_i^C \leq H_2, \forall i \in V_2, \tag{16}$$

$$\sum_{(i,j)\in A_1} x_{i,j}^k \times F_{i,j}^D \leq E_1, \forall k \in K_1, \tag{17}$$

$$\sum_{(i,j)\in A_2} x_{i,j}^k \times F_{i,j}^C \leq E_2, \forall k \in K_2, \tag{18}$$

$$x_{i,j}^k \in \{0,1\}, k \in K, (i,j) \in A. \tag{19}$$

The objective function (10) is to minimize the total collection time of $f_1$ and exposure index of $f_2$. Constraints (11) require that each deliverer drone starts from and returns to the testing center. Constraints (12) require that the starting and ending points of each collector drone must be at the transit points. Constraints (13) are flow conservation constraints. Constraints (14) guarantee that each user point is traversed by a collector only once. Constraints (15) and (16) ensure that the sample load does not exceed the maximum capacity of the drone. Constraints (17) and (18) are energy constraints that ensure the drone can return to the testing center or transit point before its energy is exhausted. Constraints (19) represent the decision variable constraints.

## 5. ALNS-RD

To minimize the exposure risk and delivery delay of infection sample collection, we integrate the ALNS-RD algorithm into the smart contract. The algorithm is executed automatically at the beginning of the BISC to obtain near-optimal flight route solutions for drones. The solution includes path sequences for collector drones to collect samples from user points and path sequences for deliverer drones to transfer samples from transit points to the testing center. The ALNS algorithm was initially proposed by Ropke and Pisinger [46], allowing for the use of different destroy and repair operators during the searching process. Based on ALNS, Rifai et al. [47] proposed the Multi-Objective Adaptive Large Neighborhood Search (MOALNS) algorithm to provide a good set of non-dominated solutions for multi-objective problems. However, the above algorithms are not suitable for solving 2E-HDRP-TS, which has strict constraints for two-echelon synchronization. Moreover, MOALNS simply determines the solution to be modified by randomly selecting a non-dominated solution from the archive, which is not conducive to the solution of 2E-HDRP-TS. Thus, this study defines four operators (i.e., flight distance-based destroy operator, flight distance-based repair operator, exposure index-based destroy operator, exposure index-based repair operator) for 2E-HDRP-TS to optimize the route of drones and improve the synchronization performance of handover operations. In addition, in order to ensure the diversity of the solutions, we set the solution with high crowding distances as the current solution for the next iteration.

### 5.1. ALNS-RD Framework

The principle of ALNS-RD is described in detail in Algorithm 1. ALNS-RD begins by generating a solution and ultimately obtains the Pareto set, in which each member represents a flight route solution for drones. The steps for each iteration of ALNS-RD are as follows: During the first iteration, $s$ is added to $S^*$. In the subsequent iterations, the algorithm continuously executes a looping process. First, the roulette wheel is used to select operators; $s$ is removed and inserted to generate $s'$; if $s'$ meets the acceptance criterion, then $s'$ is chosen as the $s$ for the next iteration; $S^*$ and the operator weights are updated; $s$ is replaced based on the crowding distance every $\delta$ iterations. Finally, we obtain the Pareto solution set $S^*$. ALNS-RD utilizes two types of operators: destroy $\phi^-$ and repair $\phi^+$ operators. The two types of operators iteratively destroy and repair the flight route solution for drones $s$ to obtain a new solution $s'$. The destroying refers to removing a number $p$ of user points from the current solution $s$, while repairing refers to inserting all pending user points in the solution. All operators initially have the same weight, which determines their probability of being selected at each iteration. The weights are updated based on the operators' historical performance. If $s'$ satisfies the acceptance criterion, it is accepted and becomes the current solution $s \leftarrow s'$ for the next iteration. Additionally, if $s'$

is non-dominated, it is inserted into the Pareto set. The acceptance criterion and the update of the Pareto set are described in detail in Section 5.7. This procedure continues until the stop-criterion is met.

---

**Algorithm 1:** ALNS-RD

---

**Input:** Current solution $s$, destroy operators $\phi^+$, repair operators $\phi^-$
**Output:** Pareto set $S^*$
**if** first iteration **then**
  $\mid$ $S^* \leftarrow s$;
**end**
**while** stop-criterion not met **do**
  $\mid$ roulette wheel selection operator;
  $\mid$ remove a certain number of user points from $s$ and insertion;
  $\mid$ generate $s'$;
  $\mid$ **if** $s'$ satisfies acceptance criterion **then**
  $\mid$ $\mid$ $s$ of next iteration $\leftarrow s'$;
  $\mid$ **end**
  $\mid$ update $S^*$;
  $\mid$ update operator weights;
  $\mid$ every $\delta$ iterations: replace $s$ based on crowding distances;
**end**
return: $S^*$

---

### 5.2. Initial Solution

We propose using a two-phase construction algorithm to obtain the initial solution, as described in Algorithm 2. Firstly, a best insertion heuristic is used to generate routes for deliverer drones to visit all used transit points, as described in lines 3–7. In this heuristic, the closest transit points in terms of distance are inserted into the routes of drones. Then, based on the deliverer drone's route, multiple routes of collector drones are generated to visit user points, as described in lines 8–12. Similarly, the best insertion heuristic is used to insert user points into existing routes of drones.

---

**Algorithm 2:** Procedure for generating initial solutions

---

**Input:** User points $V_u$, transit point $V_s$, number of deliverer drones $|K_1|$, number of collector drones $|K_2|$
**Output:** Initial solution
Initialize the deliverer route, starting from the TC;
Initialize the collector routes, starting from a random transit point;
**while** unvisited transit points exist **do**
  $\mid$ Find the transit point $s$ closest to the last point on the route;
  $\mid$ Insert end of route;
  $\mid$ Add the starting city to the end of the route;
**end**
**while** unvisited user points exist **do**
  $\mid$ Find the user point $u$ closest to the last point on the route;
  $\mid$ Insert end of route;
  $\mid$ Add the starting city to the end of the route;
**end**
return: Initial solution

---

### 5.3. Non-Dominated Solution

2E-HDRP-TS is a multi-objective problem, which requires minimization of the total completion time of sample collection and the exposure index of samples during the han-

dover process. In most cases, these two objectives are in conflict. When the total collection time of infection samples is prioritized, the vast majority of samples are likely to be sent to the closest transit point to the testing center to ensure that the entire sample collection task can be completed quickly. However, considering that the completion time of the handover mainly depends on the last drone to arrive, doing so may result in these samples staying at this transit point for too long, which in turn increases the risk of sample exposure. This conflict between objectives prevents us from obtaining a dominant optimal solution. Therefore, we generate a Pareto solution set $S^*$ to preserve the various non-dominated solutions $s$. In the 2E-HDRP-TS, the Pareto dominance relation can be defined when a solution $s \in S^*$ dominates a solution $s'$, denoted as $s \succ s'$, if and only if the following conditions are fulfilled:

$$f_O(s) \leq f_o(s'), \forall o \in O, \tag{20}$$

$$f_O(s) < f_o(s'), \exists o \in O, \tag{21}$$

where $O$ is the set of optimization objectives. Equations (20) and (21) indicate that $s$ can be equal to $s'$ on one objective, but another objective must be better than $s'$. During the iteration process, $s$ will be inserted into the Pareto set $S^*$. When a new solution that dominates $s$ appears, $s$ will be removed from the Pareto set $S^*$. The Pareto front of $S^*$ moves forward with the optimization and is closer to the true Pareto front.

### 5.4. Destroy Operators

We generate a new solution by destroying and repairing the current flight route solution for drones. This process assigns infection samples to different collector drones and replans their visit routes. To destroy a current solution, we first select an operator and continuously remove user points until the removal requirement is met. There are two types of destroy operators: route-based destroy operators and user-based destroy operators.

#### 5.4.1. Route-Based Destroy Operators

Route-based destroy operators remove the entire route of drones and contain the following three operators.

- Random route removal: For the current solution, this operator randomly removes a route of the drone, which means removing all user points on this route.
- Random drone removal: This operator randomly removes all user points collected by the drone. It is hoped that the repair operator will assign uncollected user points to other drones. Random drone removal is a more effective method than random route removal.
- Exposure index-based route removal: Based on the low-exposure requirement of the two-echelon infection sample collection with synchronization, we proposed a destroy operator based on the exposure index. When the drones participating in the handover arrive almost synchronously, the exposure index at this handover naturally is very low. If the collector drone arrives a long time before (or after) the deliverer drone, the handover operation will involve a long waiting time and will pose a threat to the sample's safety. We are inspired by the worst removal [46] and directly remove the poorly performing routes in the handover. The operator begins by calculating the exposure index of each handover operation. After sorting, all user points on the worst route are removed until the removal requirement is met. This operator removes routes with poor synchronization performance to improve the safety of the sample.

#### 5.4.2. User-Based Destroy Operators

User-based destroy operators remove certain user points from the drone's route and contain the following four operators.

- Random user point removal: This operator randomly removes user points from the drone's route until the removal requirement is met.
- Worst user point removal: For each user point, this operator calculates the difference between the cost function with and without this user point. The cost function repre-

sents the distance required to collect samples at that point. Then, the operator sorts the user points from the highest to the lowest cost and removes the worst user points from the drone's route.

- Related user point removal: This operator is based on the Shaw removal heuristic [48], and it deletes a certain number of related user points from the drone's route. We measure the relatedness between two user points based on their distance. A shorter distance indicates a great degree of relatedness. This operator is designed to delete closely related user points.
- Flight distance-based removal: To minimize the total collection time of infection samples, we design a flight distance-based destroy operator.

As shown in Figure 9, the flight distance of the sample at user point 1 follows the route of the collector drone to transit point 4 first, and then follows the route of the deliverer drone to the testing center (node 0). In order to evaluate the potential impact of an individual user point on the objective value $T$, the following analysis is conducted. First, according to the flight time of the collector drone in Equation (3), we calculate the time increment $t_1$ for user point 1 on this route:

$$t_1 = \frac{q * D_{1,4}^F}{370\eta\gamma(P_2 - e)},$$ (22)

where $q$ is the quality of a single sample. $D_{1,4}^F$ is the flight distance from user point 1 to transit point 4 along the route. The other parts of Equation (22) can be considered as constants when the drone's parameters are determined. Therefore, $t_1$ is proportional to $D_{1,4}^F$ but independent of the drone's weight and total payload.

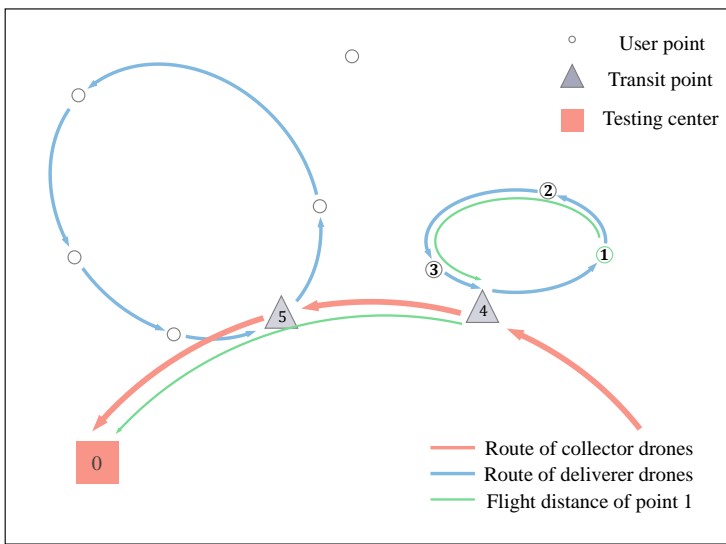

**Figure 9.** Flight distance of sample at user point 1.

Figure 10 further explains the relationship between the flight time and flight distance of the user point. The $x$ axis represents the drone's weight and the quality of each user point's sample $q$ ($w_2 > q$). The drone's route from the starting point passes through each user point, and finally returns to the endpoint. The size of each column rectangle represents the specific time increment for the corresponding weight portion in this route.

The Equation (22) can also be refined by considering the flight distance in the route of the deliverer drone. We assume that the handover operation is perfectly synchronized, meaning there is no waiting time for both drones involved in the handover link. The time increment of user point 1 with respect to the objective value $T$ is given as

$$t_1' = \frac{q * D_{1,4}^F}{370\eta\gamma(P_2 - e)} + \frac{q * D_{4,0}^F}{370\eta\gamma(P_1 - e)}.$$ (23)

In summary, there are two methods that can be used to shorten the collection time when the sample quality is fixed. The first method is to optimize the route and find shorter circuits. The second method is to find and remove user points with large time increments from the flight route and assign them to drones with lower collection costs. Since the time increment is proportional to the flight distance of the user point, we propose a flight distance-based destroy operator. The operator aims to remove user points with the longest flight distances. Instead of calculating the difference in objective value with and without these user points, it calculates the increment in the objective value in terms of flight distance. Removing the user point with the farthest flight distance after sorting can significantly reduce the total collection time of infection samples.

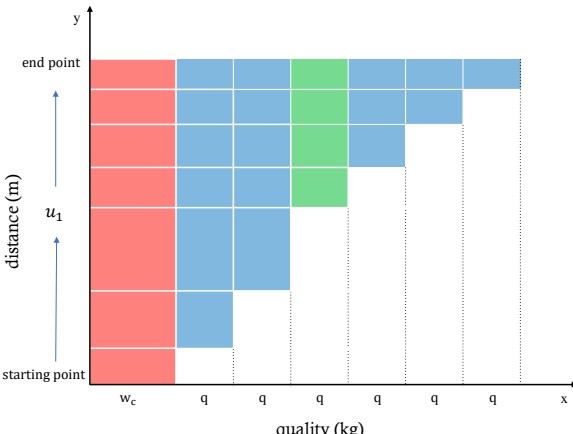

**Figure 10.** The relationship between time increment and flight distance: The red part represents the time increment caused by the drone's self-weight. The green part represents the time increment caused by user point 1, which is the $t_1$. The blue part represents the time increment caused by the drone when collecting samples from other user points.

### 5.5. Repair Operators

In Section 5.4, we remove some user points from the drone's routes in order to generate a new solution, which results in some infection samples being uncollected in the solution. Therefore, we use the repair operator to insert the uncollected user points into the drone's route. In the repair process, the insertion can either be conducted along the existing route or another route can be created for the drone. User-based destroy operators contain the following four operators.

- Greedy insertion: For an uncollected user point, after traversing all possible insertion positions, this operator chooses the position with the lowest cost for insertion.
- Random insertion: This operator randomly selects insertion positions and chooses the most suitable position among them to insert the uncollected user points.
- Flight distance-based insertion: Similar to the flight distance-based destroy operator, we propose a flight distance-based repair operator. This operator traverses each route and selects the most suitable insertion position based on the minimum flight distance principle.
- Exposure index-based insertion: This operator is aimed at routes that still have room for improvement in terms of exposure index. Uncollected user points are continuously inserted at suitable positions along the route to reduce the waiting time for drones in handover operations. During this process, the selected route keeps expanding until it reaches saturation, where both drones arrive at the transit point almost synchronously.

### 5.6. Adaptive Mechanism

The ALNS-RD algorithm utilizes an adaptive mechanism to select multiple destroy and repair operators. The adaptive mechanism can adjust throughout the entire iteration process to better adapt to different situations. The algorithm adopts a roulette wheel strategy to calculate the probability of different operators being selected. At the beginning

of the iteration, the probability of selecting each operator is the same. Each operator is assigned a dynamic weight $\omega$, which depends on its contribution to the objective value. The weights of the repair operators $\phi^+$ and the destruction operators $\phi^-$ do not interfere with each other. Specifically, we evaluate the performance of the operators in previous iterations by their scores, and update their weights. Finally, the selection probability of each operator is calculated based on the weights and shown as

$$p_{\phi^-}(j) = \omega_j / \sum_{i \in \phi^-} \omega_i, \forall j \in \phi^-, \qquad (24)$$

$$p_{\phi^+}(j) = \omega_j / \sum_{i \in \phi^+} \omega_i, \forall j \in \phi^+. \qquad (25)$$

At the beginning of the iteration, all operators are initialized and have an equal probability of regenerating routes for drones. After entering the iteration, based on the score $\pi_{i,j}$ obtained by the operator $i$ in the $j$-th loop, the weight of selecting the operator $i$ in the next loop is updated as $\omega_{i,j+1}$. The score $\pi_{i,j}$ represents the performance of the operator $i$ in the $j$-th iteration, and $\mu_{i,j}$ is the number of times that the operator $i$ was used in the $j$-th iteration. When the current iteration ends, the scores obtained by the operators are used to calculate the new weights. The weight update process is shown as follows:

$$\omega_{i,j+1} = \begin{cases} \omega_{i,j} & if \ \mu_{i,j} = 0 \\ (1-r)\omega_{i,j} + r\pi_{i,j}/\mu_{i,j} & if \ \mu_{i,j} \neq 0, \end{cases} \qquad (26)$$

where $r \in [0,1]$ is the response factor. The $r$ controls the sensitivity of the adaptive selection of the algorithm and ranges from 0 to 1. As $r$ approaches one, the new weights essentially depend on the score of this iteration. The parameter $\pi_{i,j}$ reflects the performance of the selected destroy and repair operators in the current iteration in the form of a score. The calculation of the score is as follows:

$$\pi_{i,j} = \max \begin{cases} \theta_1 & if \ s \succ s^P, \ \exists \ s^P \in S^* \\ \theta_2 & if \ s \succ s^P \ or \ s^P \succ s, \ \nexists \ s^P \in S^* \\ \theta_3 & if \ s' \succ s \\ \theta_4 & if \ s' \ \text{is accepted as a suboptimal solution} \\ \theta_5 & if \ s' \ \text{not accepted}, \end{cases} \qquad (27)$$

where $s'$ represents the new solution generated by destroying and repairing a current solution $s$. $s^P$ is a solution of the Pareto set $S^*$. The acceptance criteria will be detailed in the next section.

### 5.7. Acceptance Criteria

The acceptance criteria of ALNS-RD are developed based on the Archived Multi-Objective Simulated Annealing (AMOSA) [49]. In AMOSA, the acceptance probability of a new solution becoming the current solution is determined by considering the dominant relationship between the new solution, the current solution, and the solutions in the archive. We propose similar acceptance criteria.

1.  If $s'$ dominates a solution in $S^*$, $s'$ will be accepted as the current solution and inserted into $S^*$. At the same time, all solutions in $S^*$ dominated by $s'$ will be deleted. The score of the operator in this situation is $\theta_1$.
2.  If $s'$ and a solution in $S^*$ are non-dominating, $s'$ will also be accepted as the current solution and will become a new Pareto solution. The score of the operator in this situation is $\theta_2$.
3.  If $s'$ dominates the current solution $s$, although $s'$ will be accepted as the current solution, it cannot be added to the Pareto set $S^*$. The score of the operator in this situation is $\theta_3$.

4.  If $s'$ and the current solution $s$ are non-dominated, whether to accept the new solution $s'$ is determined by the difference with the Pareto solution set $S^*$. The closer the distance to the Pareto set $S^*$, the higher the probability of being accepted. In this case, the amount of domination is used to measure the difference between two solutions [49]. Between two solutions $a$ and $b$, the amount of domination is defined as

$$\triangle_{a,b} = \prod_{o=1, f_o(a) \neq f_o(b)}^{O} \left( \frac{|f_o(a) - f_o(b)|}{R_o} \right), \tag{28}$$

where $R_o$ represents the range of the objective $o \in O$, obtained using Pareto set $S^*$. When $s'$ is dominated by $m$ solutions in set $S^*$, the average amount of domination can be defined as $\triangle_A = (\sum_{i=1}^{k} \triangle_{i,s'})/m$. And we pick $s'$ as the current solution with probability $p_C$, which is defined as $p_C = 1/(1 + e^{\triangle_A * T_{SA}})$. $T_{SA}$ represents the current temperature, which is decreased in each iteration by being multiplied by the cooling rate $c$. The score of the operator is $\theta_4$ if the solution is accepted and $\theta_5$ otherwise.

### 5.8. Crowding Distance and Current Solution Replacement

To prevent premature convergence to a local optimum, we select a solution from the Pareto set $S^*$ as the current solution at an appropriate time. Specifically, if a new solution meets the acceptance criterion, then it is promoted to the current solution for the next iteration. Otherwise, ALNS-RD continues to generate new solutions based on $s$ or select the point with the maximum crowding distance from the Pareto set $S^*$ as the current solution.

In the Pareto set $S^*$, crowding distance refers to the density around a point, which is the difference in objective values between the point and its neighbors. As shown in Figure 11, the crowding distance of solution $n$ can be calculated by

$$C_n = \sum_{o=1}^{O} \frac{f_o(n+1) - f_o(n-1)}{R_o}, \tag{29}$$

where $R_o$ represents the range of the objective $o \in O$ obtained using Pareto set $S^*$. If the left or right neighbors are empty, the crowding distance $C_n$ is regarded as infinitely large. A higher crowding distance typically implies a lower similarity between this solution and other solutions, indicating that the neighborhood search for this solution is not extensive enough. Consequently, if there is still no improvement after $\delta$ iterations, ALNS-RD selects the solution with the maximum crowding distance as the current solution for the next iteration. The diversity of the Pareto set is maintained in this way.

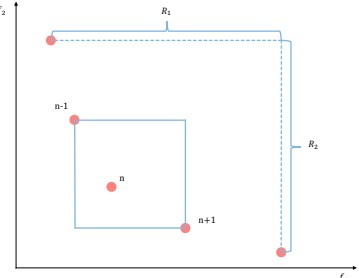

**Figure 11.** Calculation of crowding distances in Pareto set.

## 6. Experimental Results

In this section, we first describe the instances used for the experiments. In Section 5.2, we introduce two performance metrics used to measure the quality of the solution set. In Section 5.3, we present the actual generated routes. The impact of ALNS-RD parameters on the solutions is discussed in Section 5.4. In Section 5.5, we discuss the impact of operators on the quality of solutions. In Section 5.6, we conduct algorithm comparisons.

### 6.1. Experimental Setup

We use Pycharm as the simulation platform and run on a computer with an Intel Xeon Gold 6230R CPU (Intel, Santa Clara, CA, USA) and an NVIDIA Quadro RTX 5000 GPU (NVIDIA, Santa Clara, CA, USA). In the experiment, we generate $1500 \times 1500$ m instances by using an actual layout. The drone's flight height is set at 30 m to fly in a free space area. The value of parameters related to collector drones and deliverer drones is set according to typical ones in practical use [45]. The weight of the deliverer drone $w_1$ is set to 9 kg, the maximum capacity $H_1$ is 11 kg, the maximum energy $E_1$ is 0.8 kW·h, and the maximum power $P_1$ is 1.5 kW. The collector drone should carry a lighter load, so the weight $w_2$ is set to 2 kg, the maximum capacity $H_2$ is 2 kg, the maximum energy $E_2$ is 0.3 kW·h, and the maximum power $P_2$ is 0.5 kW. We assume that the area is covered by the 4G network and the drones are capable of maintaining real-time communication with ground facilities (user points and agencies). We assume that the communication link between the drones and ground facilities is line-of-sight communication. The communication channel modeling and parameter settings refer to [50]. All simulation parameters are presented in Table 3. In this simulation, it is assumed that one deliverer drone and four collector drones complete the collection task.

**Table 3.** Simulation parameters.

| Parameter | Value |
| --- | --- |
| The flight height | 30 m |
| The speed of a delivery drone with full payloads | 40 km/h |
| The speed of a collector drone with full payloads | 56 km/h |
| The maximum capacity of the deliverer drone $H_1$ | 11 kg |
| The maximum capacity of the collector drone $H_2$ | 2 kg |
| The maximum energy of the deliverer drone $E_1$ | 0.8 kW·h |
| The maximum energy of the collector drone $E_2$ | 0.3 kW·h |
| The weight of the deliverer drone $w_1$ | 9 kg |
| The weight of the collector drone $w_2$ | 2 kg |
| The maximum power of the deliverer drone $P_1$ | 1.5 kW |
| The maximum power of the collector drone $P_2$ | 0.5 kW |
| The sample quality of user points $q$ | 0.1 kg |
| The power transfer efficiency for motor and propeller $\eta$ | 0.5 |
| The lift-to-drag ratio $\gamma$ | 3 |
| The energy consumption of other electronic devices on the drone $e$ | 0.1 kW |
| The carrier frequency $f_c$ | 2 GHz |
| Channel parameter $\eta_{\text{LoS}}$ | 3 dB |
| Channel environmental parameter $\psi$ | 9.61 |
| Channel environmental parameter $\beta$ | 0.16 |

### 6.2. Instance Description

We first generate instances used for the experiments and then present the geographical configuration of the BISC used in the instances.

#### 6.2.1. Generating Instance

We generate instances by using an actual layout of residential users near the countryside of Ekron, Kentucky, United States. As shown in Figure 12, this area is located far from the city and is suitable for two-echelon sample collection with synchronization. Additionally, there are no buildings in the countryside to obstruct drone flights, so the route of the drone does not need to consider obstacle avoidance. The drone flies between user points and transit points, delivering all infection samples from the user points to the testing center.

Specifically, we obtain the real geographic locations through Google Earth and measure the scale of the entire scene. Then, we label the locations of the user points using ImageJ (https://imagej.net/ij/), an image analysis software program. The instances are generated

with the map origin at the bottom-left corner, and the coordinates of all user points are computed to obtain the distance matrix.

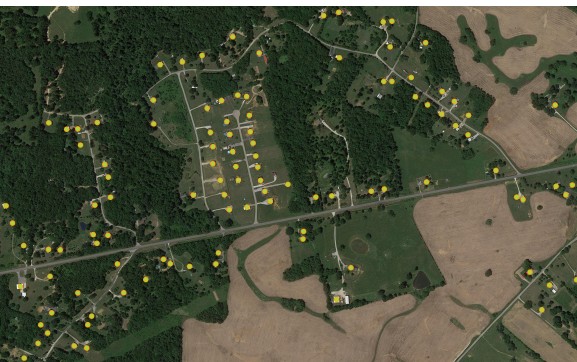

**Figure 12.** The actual layout of residential users near the countryside of Ekron (108 user points).

### 6.2.2. Geographical Configuration

Based on the geographical configuration proposed by Grangier et al. [34], we adopt the following $X/Y/M/N$ notation to describe the positions of the testing center and the transit points. $X$ and $Y$ are expressed as a percentage of the map size, indicating the location of a testing center. $M$ and $N$ represent the number of rows and columns of the grid. A transit point is located at each intersection of the grid. Figure 13 shows a $-50/50/3/3$ configuration, which is used in all experiments. Finally, we combine the actual layout of residential users and geographical configuration to create an instance called TDR1. The following route results and parameter tuning of ALNS-RD are based on TDR1.

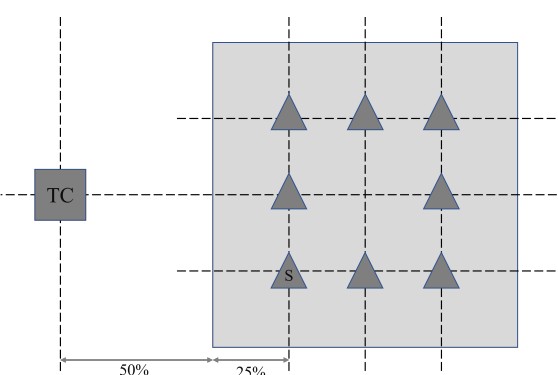

**Figure 13.** Geometrical layout.

### 6.3. Performance Metrics

To evaluate the performance of ALNS-RD, we use the following two performance metrics to measure the quality of the Pareto set.

### 6.3.1. Perfect Point

In order to evaluate the quality of the obtained solutions, we introduce the perfect point $f^b$. The perfect point $f^b$ is the best possible point, which has the lowest value in each objective of the Pareto set generated by ALNS-RD. $f^b_o$ represents the perfect value of the perfect point $f^b$ in objective $o \in O$ and is calculated by

$$f^b_o = \min_{s \in S^*} f_o(s), \forall o \in O. \tag{30}$$

where $O$ represents the set of optimization objectives, $O = \{1, 2\}$. $f^b_1$ represents the optimal total collection time of infection samples $T$ of the entire Pareto set $S^*$, and $f^b_2$ represents the optimal exposure index $Q$ of the entire Pareto set $S^*$.

### 6.3.2. Average Euclidean Distance

The average Euclidean distance is commonly used to evaluate the convergence of a solution set. The Euclidean distance from the solution $s$ to the perfect point $f^b$ is denoted as $d_s$, which can be calculated as follows

$$d_s = \sqrt{\sum_{o=1}^{O} \left( \frac{f_o(s) - f_o^b}{R_o} \right)^2}, \tag{31}$$

where $R_o$ is the range of the objective $o \in O$ in the Pareto set $S^*$. The average Euclidean distance of all solutions in the $S^*$ is denoted as $d_A$, which can be calculated as follows

$$d_A = \frac{\sum_{s \in S^*} d_s}{|S^*|}, \tag{32}$$

where $|S^*|$ is the number of solutions in the Pareto set $S^*$. A lower value of average Euclidean distance $d_A$ implies that all solutions are closer to the center of the solution space. This also means that the Pareto set $S^*$ is closer to the true Pareto front.

### 6.4. Simulated Route Results

In this section, we show the simulated route results based on instance TDR1 using ALNS-RD.

As shown in Figure 14, we select the three most representative solutions from the Pareto set for route presentation, which are the solution that minimizes the total collection time of infection samples $T$, the solution that minimizes the exposure index $Q$, and the trade-off solution. In this solution, the deliverer drone has one route, while the collection drones have two routes. Therefore, we present the routes of drones in two separate figures, with each drone represented by a different color.

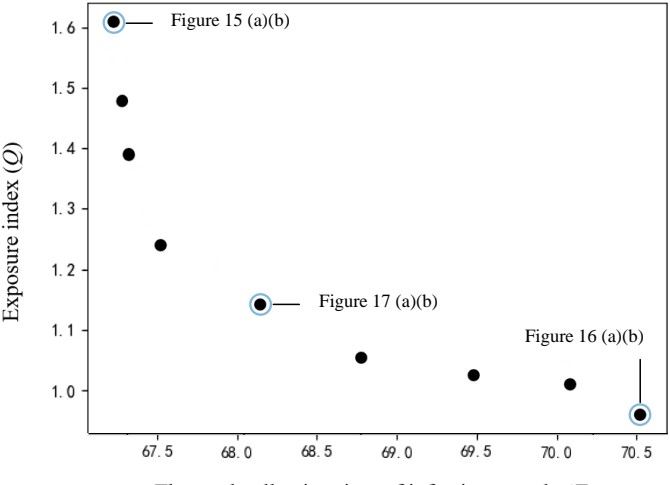

**Figure 14.** Pareto front based on instance TDR1 using ALNS-RD (Figures 15–17 describe specifically).

Figure 15 shows the flight routes of the drones presented by the solution that minimizes the total collection time of infection samples $T$. As shown in Figure 15a, the first routes of the collector drones depart from the transit point and return to the same transit point for handover, completing the collection of infection samples at each user point in the process. There is no intersection between routes of collector drones. This indicates that the drones tend to make the shortest route decision, resulting in the shortest collection time. As shown in Figure 15b, the collector drones all choose the nearest transit point to the testing center, aiming to minimize the total collection time of infection samples.

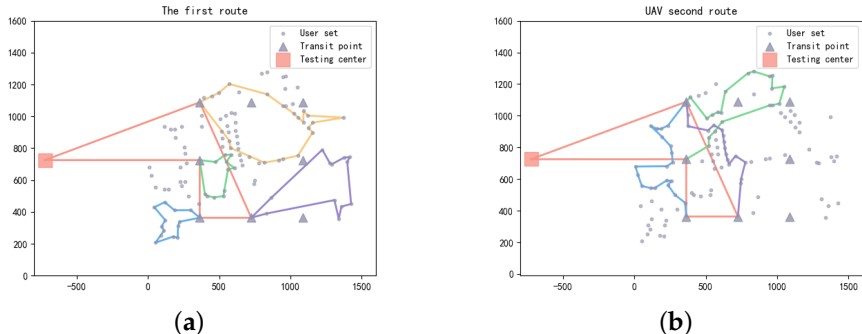

**Figure 15.** The flight routes of the solution that minimize the total collection time of infection samples *T*. (**a**) First route. (**b**) Second route.

Figure 16 shows the flight routes of the drones presented by the solution that minimizes the exposure index *Q*. In the Pareto set, to optimize the exposure index as much as possible and improve the synchronization of handover operations, the total collection time of infection samples *T* has to be increased. In this case, the solution with the shortest route may not lead to the optimal objective value *Q*. We observe that in Figure 16a,b, there are route intersections, which means that these routes are not the shortest circuits. ALNS-RD can control the arrival time of the collector drone at the transit point by extending its route, so that the collector drone synchronizes with the deliverer drone at the transit point.

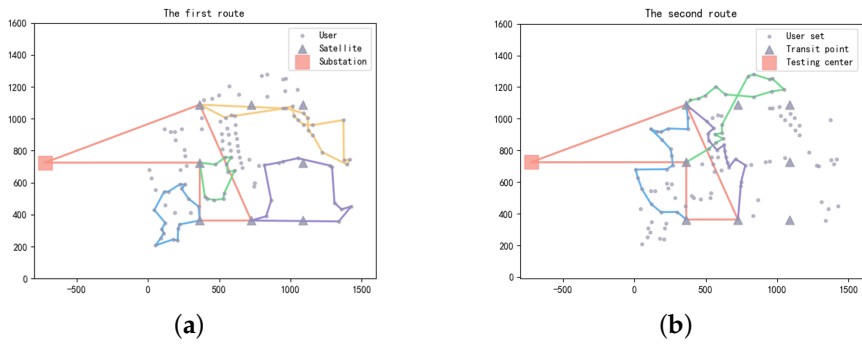

**Figure 16.** The flight routes of the solution that minimizes the exposure index *Q*. (**a**) First route. (**b**) Second route.

Figure 17 shows the flight routes of the trade-off solution, in which the drones balance the collection time and exposure index. This solution maintains a relatively low collection time of infection samples and the exposure index. This trade-off ensures efficient collection while keeping the risk of infection spread acceptable. Although this solution may not be optimal, it is highly possible for it to be chosen in practical applications.

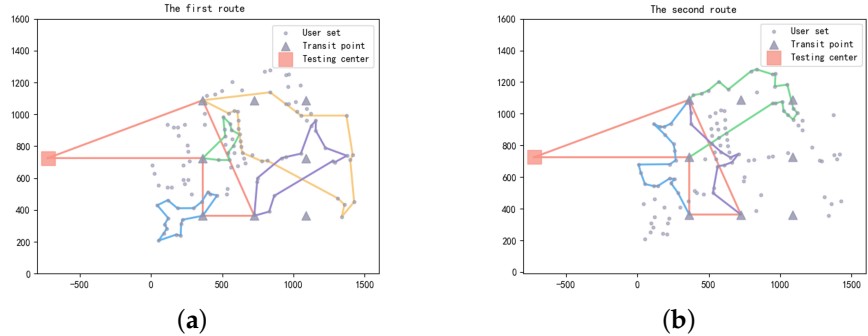

**Figure 17.** The flight routes of the trade-off solution. (**a**) First route. (**b**) Second route.

Figure 18 shows the payload variation of the drones during the collection and delivery process in the trade-off solution. Collector drones 1, 2, and 3 each experience two payload drops, resulting in two transit operations. Collector drone 4 visits numerous user points located on the edges and performs one transit operation.

In real applications, the Pareto set can provide multiple alternative solutions. The decision maker can select a flight route solution for drones from the Pareto set that best fulfills requirements.

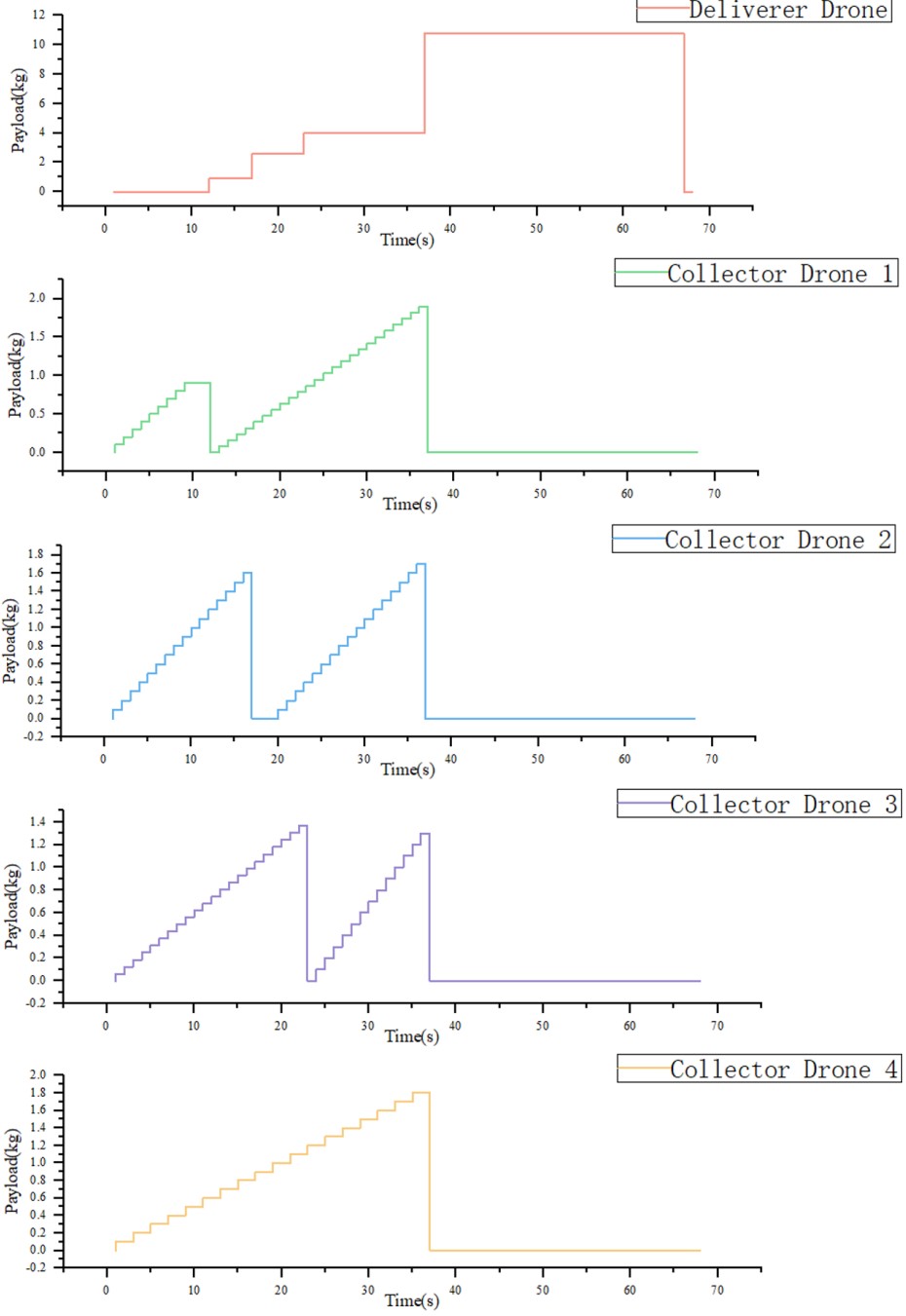

**Figure 18.** The payload variation of the drones during the collection and delivery process in the trade-off solution.

*6.5. Parameter Tuning*

In this section, we describe how the ALNS-RD parameters are used in the following experiments.

Despite the large number of parameters used in the ALNS-RD, it turns out that it is possible to find a set of parameters that works well for multiple instances. We use the following strategy to tune the parameters. We first generate a set of fair parameter settings. Then, we allow a particular key parameter to take a number of values while the rest of the parameters are kept fixed. With an insufficient number of iterations (1000), the perfect value $f_o^b$ will be affected by the change of parameter values. Insufficient iterations can amplify the differences caused by different parameter values. For each parameter setting, the perfect values $f_o^b$ are calculated ten times, and the average is taken. Deviation refers to the difference between the average perfect value and the best perfect value in the entire experiment, which is used to evaluate the rationality of the experimental parameter settings. The lower the deviation, the more likely it is that the solution set obtained based on this parameter is close to the true Pareto front. We choose the settings with the best average perfect value and move on to the next parameter. This process continues until all parameter values have been determined.

First, we determine the value of the destroy ratio $\alpha$ ($\alpha = p/|V_u|$). It represents the ratio of the number of user points removed to the total number of user points. Generally, the removal rate $\alpha$ has a significant impact on the convergence speed of ALNS-RD and the objectives. The removal rate $\alpha$ ranges from 0.05 to 0.5, with a step size of 0.05. Table 4 shows the impact of $\alpha$ on perfect value $f_o^b$. Obviously, $\alpha$ ranges from 0.2 to 0.3 is a good choice and we choose 0.3.

**Table 4.** Impact of $\alpha$ on perfect value $f_o^b$.

| Destroy Ratio $\alpha$ | 0.10 | 0.15 | 0.20 | 0.25 | 0.30 | 0.35 | 0.40 | 0.45 | 0.50 |
|---|---|---|---|---|---|---|---|---|---|
| Dev. $f_1^b$ (%) | 20.1 | 28.8 | 8.9 | 9.6 | 10.4 | 10.8 | 11.7 | 11.8 | 10.9 |
| Dev. $f_2^b$ (%) | 92.6 | 109.3 | 50.7 | 51.2 | 42.5 | 71.9 | 72.1 | 74.5 | 80.0 |

The reaction coefficient $r$ is another key parameter that controls the sensitivity of the adaptive selection operator during the weight update process. This parameter is set in the range of 0.1 to 0.8 with a step size of 0.1. By applying the same parameter tuning strategy, the results in Table 5 are obtained. The results show that the deviation of average perfect value is minimized when $r$ is set between 0.4 and 0.5. Therefore, we set the reaction coefficient $r$ to 0.5.

**Table 5.** Impact of $r$ on perfect value $f_o^b$.

| Reaction Coefficient $r$ | 0.10 | 0.20 | 0.30 | 0.40 | 0.50 | 0.60 | 0.70 | 0.80 |
|---|---|---|---|---|---|---|---|---|
| Dev. $f_1^b$ (%) | 11.9 | 12.3 | 11.5 | 7.9 | 8.9 | 10.2 | 9.1 | 12.1 |
| Dev. $f_2^b$ (%) | 97.3 | 75.8 | 77.0 | 71.5 | 61.0 | 72.3 | 74.3 | 82.8 |

In addition to the damage rate $\alpha$ and the reaction coefficient $r$, the ALNS-RD algorithm includes the following parameters $(c, \theta_1, \theta_2, \theta_3, \theta_4, \theta_5, \delta)$. Referring to the study of Ropke & Pisinger [46], the values of $(c, \theta_1, \theta_2\ \theta_3\ \theta_4\ \theta_5, \delta)$ are taken as (0.99975, 20, 16, 10, 6, 2, 10), respectively.

*6.6. Impact of Operators on Solutions Quality*

In this section, we discuss the impact of neighborhoods composed of different operators on the quality of solution sets. To make the experimental results convincing enough, we run multiple neighborhoods on instances of different scales. Table 6 provides detailed descriptions of the instances (TDR1-TDR5).

**Table 6.** Instances information, $|K_1|$ represents the number of deliverer drones, $|K_2|$ represents the number of collector drones and TC represents the testing center.

| Instance | Range | Scale | $|K_1|$ | $|K_2|$ | Geographical Configuration |
|----------|-------|-------|---------|---------|----------------------------|
| TDR1 | 1500 m × 1500 m | 108 user points | 1 | 4 | 1 TC and 4 transit points |
| TDR2 | 1500 m × 1500 m | 108 user points | 1 | 4 | 1 TC and 4 transit points |
| TDR3 | 1500 m × 1500 m | 102 user points | 1 | 4 | 1 TC and 4 transit points |
| TDR4 | 1500 m × 1500 m | 60 user points | 1 | 4 | 1 TC and 4 transit points |
| TDR5 | 1500 m × 1500 m | 50 user points | 1 | 4 | 1 TC and 4 transit points |

We combine different destroy and repair operators, as shown in Table 7. Neighborhood 1 consists of the basic operators mentioned in Sections 5.4 and 5.5. Neighborhood 2 incorporates flight distance-based destroy and repair operators in addition to those in Neighborhood 1. Neighborhood 3 further includes exposure index-based destroy and repair operators. We perform insufficient iterations (1000) for the convergence speed and solution quality of different neighborhoods. We record the best perfect values from 10 runs and the average perfect value. Deviation refers to the difference between the average perfect value in the neighborhood and the best perfect value in the entire experiment. The results are presented in Tables 8 and 9.

**Table 7.** Neighborhood configuration.

| Neighborhood | Basic | Flight Distance-Based | Exposure Index-Based |
|--------------|-------|-----------------------|----------------------|
| Neighborhood 1 | ✓ | | |
| Neighborhood 2 | ✓ | ✓ | |
| Neighborhood 3 | ✓ | ✓ | ✓ |

**Table 8.** Comparison of the perfect value $f_1^b$ over 10 runs for 3 different neighborhood configurations (The best performance metrics are bolded).

| | Neighborhood 1 | | | Neighborhood 2 | | | Neighborhood 3 | | |
|---------|------------------|-----------------|---------------------|------------------|-----------------|---------------------|------------------|-----------------|---------------------|
| Instance | Best.$f_1^b$ | Avg.$f_1^b$ | Dev.$f_1^b$(%) | Best.$f_1^b$ | Avg.$f_1^b$ | Dev.$f_1^b$(%) | Best.$f_1^b$ | Avg.$f_1^b$ | Dev.$f_1^b$(%) |
| TDR1 | 73.5 | 78.0 | 15.6 | 68.7 | 72.0 | 7.6 | **67.5** | **71.9** | **6.5** |
| TDR2 | 78.0 | 80.4 | 15.5 | **67.6** | **71.0** | **5.0** | 68.5 | 71.6 | 5.8 |
| TDR3 | 75.1 | 79.0 | 17.9 | **66.9** | **71.5** | **6.8** | 67.0 | 73.0 | 8.8 |
| TDR4 | 66.9 | 69.8 | 22.7 | 57.3 | 61.4 | 7.9 | **56.9** | **60.3** | **6.0** |
| TDR5 | 61.1 | 62.5 | 17.6 | **54.4** | 58.9 | 8.9 | 8.2 | **57.8** | **6.1** |

**Table 9.** Comparison of the perfect value $f_2^b$ over 10 runs for 3 different neighborhood configurations (The best performance metrics are bolded).

| | Neighborhood 1 | | | Neighborhood 2 | | | Neighborhood 3 | | |
|---------|------------------|-----------------|---------------------|------------------|-----------------|---------------------|------------------|-----------------|---------------------|
| Instance | Best.$f_2^b$ | Avg.$f_2^b$ | Dev.$f_2^b$(%) | Best.$f_2^b$ | Avg.$f_2^b$ | Dev.$f_2^b$(%) | Best.$f_2^b$ | Avg.$f_2^b$ | Dev.$f_2^b$(%) |
| TDR1 | 1.61 | 2.00 | 70.6 | 1.23 | 1.51 | 29.2 | **1.17** | **1.42** | **21.0** |
| TDR2 | 5.18 | 6.20 | 63.4 | 4.10 | 4.73 | 24.4 | **3.83** | **4.23** | **12.5** |
| TDR3 | 5.63 | 6.41 | 64.0 | 4.67 | 5.11 | 30.8 | **3.91** | **4.67** | **19.4** |
| TDR4 | 4.98 | 5.88 | 78.8 | 3.80 | 4.54 | 37.9 | **3.29** | **3.75** | **13.9** |
| TDR5 | 2.94 | 3.47 | 115.8 | 1.96 | 2.20 | 36.8 | **1.61** | **1.81** | **12.4** |

Tables 8 and 9 show the impact of the different neighborhoods on the solution quality. We observe that the flight distance-based operator is highly effective at optimizing the total collection time of infection samples $T$. Neighborhood 2 incorporates this operator into the search process and significantly improves the quality of the objective $T$. However, we found that when the exposure index-based operator is used to optimize the total collection

time of infection samples $T$, its impact on the quality of the Pareto set is very limited. Specifically, in Table 8, we can observe that the results obtained from Neighborhood 2 and Neighborhood 3 are very close. In contrast, in Table 9, the best perfect value and average perfect value for neighborhood 3 with the exposure index-based operator are improved significantly. In general, the flight distance-based operators and the exposure index-based operators play different roles in minimizing the objective value. The former is very effective at optimizing the total collection time of infection sample $T$, while the latter can significantly improve the exposure index $Q$. The results prove that the proposed flight distance-based operators and the exposure index-based operators can generate a Pareto set that is closer to the true Pareto front. Based on the experimental results, we configure neighborhood 3 for algorithm comparison.

### 6.7. Algorithm Comparison

In this section, we demonstrate the effectiveness of the proposed ALNS-RD algorithm by comparing it with benchmark algorithms (i.e., NSGA-II [51], AMOSA [49], MOLNS [52] and MOALNS [47]). NSGA-II (Non-dominated Sorting Genetic Algorithm II) is commonly used in multi-objective optimization and is one of the classic multi-objective optimization algorithms. In addition, MOLNS (Multi-Objective Large Neighborhood Search), which is similar to ALNS, is also included to evaluate the significance of the improvement. All algorithms are run using the same equipment and software to avoid the bias. The ALNS-RD algorithm is configured with a neighborhood 3 and iterated 10,000 times. The MOLNS, MOALNS and AMOSA follow the same parameter settings as ALNS-RD to ensure the same number of iterations. As NSGA-II is a population-based optimization, the number of iterations is set differently compared to other algorithms. The parameters in NSGA-II were configured using the recommended parameters in [51], which are shown in Table 10.

**Table 10.** Parameter settings of NSGA-II.

| Parameter | Value |
|---|---|
| Maximum generation | 1400 |
| Population size | 20 |
| Crossover probability | 0.9 |
| Mutation probability | 0.2 |

As shown in Tables 11 and 12, for the Pareto sets obtained by running the different algorithms ten times, we calculate the best perfect value $Best.f_o^b$, the average Euclidean distance of the Pareto Set with $Best.f_o^b$, the average perfect value $Avg.f_o^b$, and the deviation. Except for TDR4, the Pareto set obtained using ALNS-RD significantly outperforms those obtained using other algorithms in all performance metrics. ALNS-RD consistently finds the Pareto set which has the best perfect value $f_1^b$ in the entire experiment. This may be due to the fact that the proposed operators provide directions for the neighborhood search, making the solution set of good quality. In particular, in instances with more than one hundred user points, the deviation of the perfect value and average Euclidean distance of the Pareto set obtained using ALNS-RD are significantly better than those obtained by other algorithms. This indicates that ALNS-RD is able to find solutions that are closer to the true Pareto front in instances with a high level of complexity. This is a highly desirable characteristic, which means that ALNS-RD is more reliable when it comes to exploring the solution space and discovering high-quality solutions. The above results show that for 2E-HDRP-TS, the ALNS-RD has good performance and generates solution sets closer to the true Pareto front.

**Table 11.** Comparison of the perfect value $f_1^b$ over 10 runs for different algorithms.

| Instance | ALNS-RD | | | | NSGA-II | | | | MOLNS | | | | MOALNS | | | | AMOSA | | | |
|---|---|---|---|---|---|---|---|---|---|---|---|---|---|---|---|---|---|---|---|---|
| | **Best.**$f_1^b$ | $d_A$ | **Avg.**$f_1^b$ | **Dev.**$f_1^b$**(%)** | **Best.**$f_1^b$ | $d_A$ | **Avg.**$f_1^b$ | **Dev.**$f_1^b$**(%)** | **Best.**$f_1^b$ | $d_A$ | **Avg.**$f_1^b$ | **Dev.**$f_1^b$**(%)** | **Best.**$f_1^b$ | $d_A$ | **Avg.**$f_1^b$ | **Dev.**$f_1^b$**(%)** | **Best.**$f_1^b$ | $d_A$ | **Avg.**$f_1^b$ | **Dev.**$f_1^b$**(%)** |
| TDR1 | 65.46 | 0.78 | 66.99 | 2.3 | 70.51 | 0.68 | 76.74 | 17.2 | 66.33 | 0.84 | 69.45 | 6.1 | 66.06 | 0.84 | 68.58 | 4.8 | 74.61 | 0.89 | 79.27 | 21.1 |
| TDR2 | 74.55 | 0.87 | 76.37 | 2.4 | 84.01 | 0.48 | 89.69 | 20.3 | 78.0 | 0.88 | 82.70 | 6.0 | 75.52 | 0.84 | 78.58 | 5.4 | 87.08 | 0.90 | 91.01 | 22.1 |
| TDR3 | 77.21 | 0.67 | 78.84 | 2.1 | 82.37 | 0.89 | 87.95 | 13.9 | 80.2 | 0.94 | 83.90 | 8.7 | 79.19 | 0.78 | 82.93 | 7.4 | 85.81 | 0.93 | 88.74 | 14.9 |
| TDR4 | 54.96 | 0.78 | 55.46 | 0.9 | 62.95 | 0.66 | 67.22 | 22.3 | 57.6 | 0.70 | 60.14 | 9.4 | 56.70 | 0.75 | 59.53 | 8.3 | 66.86 | 0.81 | 70.06 | 27.5 |
| TDR5 | 53.56 | 0.74 | 53.91 | 0.7 | 58.05 | 0.77 | 62.43 | 16.6 | 54.66 | 0.87 | 56.75 | 6.0 | 54.10 | 0.87 | 57.13 | 6.7 | 61.63 | 0.91 | 64.98 | 21.3 |

**Table 12.** Comparison of the perfect value $f_2^b$ over 10 runs for different algorithms.

| Instance | ALNS-RD | | | | NSGA-II | | | | MOLNS | | | | MOALNS | | | | AMOSA | | | |
|---|---|---|---|---|---|---|---|---|---|---|---|---|---|---|---|---|---|---|---|---|
| | **Best.**$f_2^b$ | $d_A$ | **Avg.**$f_2^b$ | **Dev.**$f_2^b$**(%)** | **Best.**$f_2^b$ | $d_A$ | **Avg.**$f_2^b$ | **Dev.**$f_2^b$**(%)** | **Best.**$f_2^b$ | $d_A$ | **Avg.**$f_2^b$ | **Dev.**$f_2^b$**(%)** | **Best.**$f_2^b$ | $d_A$ | **Avg.**$f_2^b$ | **Dev.**$f_2^b$**(%)** | **Best.**$f_2^b$ | $d_A$ | **Avg.**$f_2^b$ | **Dev.**$f_2^b$**(%)** |
| TDR1 | 1.13 | 0.66 | 1.18 | 4.2 | 2.04 | 0.87 | 3.62 | 220.4 | 1.48 | 0.94 | 2.39 | 111.2 | 1.59 | 0.86 | 1.96 | 73.5 | 1.83 | 0.82 | 2.68 | 137.2 |
| TDR2 | 3.02 | 0.70 | 3.21 | 6.2 | 5.03 | 0.85 | 6.32 | 109.3 | 4.32 | 0.85 | 6.09 | 101.7 | 3.92 | 0.85 | 5.38 | 93.0 | 5.72 | 0.81 | 7.48 | 147.7 |
| TDR3 | 2.67 | 0.64 | 2.78 | 4.1 | 6.89 | 0.68 | 8.34 | 212.4 | 5.38 | 0.89 | 6.92 | 159.2 | 4.92 | 0.87 | 5.61 | 167.8 | 6.98 | 0.88 | 8.87 | 232.2 |
| TDR4 | 2.21 | 0.78 | 2.43 | 10.0 | 3.33 | 0.39 | 5.40 | 144.3 | 3.11 | 0.66 | 5.09 | 130.3 | 2.75 | 0.79 | 4.31 | 95.0 | 3.20 | 0.96 | 4.12 | 86.4 |
| TDR5 | 0.99 | 0.62 | 1.04 | 4.8 | 2.07 | 0.71 | 2.94 | 197.0 | 1.61 | 0.95 | 2.27 | 129.3 | 1.46 | 0.94 | 1.82 | 83.8 | 1.97 | 0.84 | 2.25 | 127.3 |

## 7. Conclusions

This study proposes a Blockchain-enabled Infection Sample Collection system (BISC) architecture, which is built with an infrastructure layer, a blockchain layer, and an application layer, in order to improve the safety and timeliness of infection sample collection. Additionally, a two-echelon drone-assisted mechanism termed the Two-Echelon Heterogeneous Drone Routing Problem with Transit point Synchronization (2E-HDRP-TS) is proposed, representing an unexplored area in this field. This problem integrates fixed transit points, synchronized handover operations, and heterogeneous drones with limited energy and capacity. The 2E-HDRP-TS can be divided into two interrelated decisions: the flight routes of collector drones for collecting infection samples from user points, and the transport routes of deliverer drones from transit points to the testing center. To generate near-optimal solutions, we introduce a multi-objective Adaptive Large Neighborhood Search algorithm for Routing of Drones (ALNS-RD), aiming to minimize the total collection time of infection samples and the exposure index. Alongside traditional search operators, the ALNS-RD algorithm incorporates innovative flight distance-based and exposure index-based operators to enhance the efficiency and accuracy of the solution. Our comparative analysis against benchmark algorithms NSGA-II, AMOSA, MOLNS, and MOALNS demonstrates the superior performance of the proposed ALNS-RD algorithm across diverse levels of complexity in all five instances.

Even though the performance of the ALNS-RD algorithm is very good, due to its nature as an offline heuristic algorithm, drones are unable to handle new task requests during the sample collection process. Hence, our future work will focus on investigating online route planning algorithms for drones, with the aim of enabling dynamic adjustment of flight routes during the sample collection and delivery process. We envision the possibility of using neural networks and reinforcement learning to develop a viable online drone route planning algorithm. In addition, Clustering users will help to complement the biological safety of infection samples. Therefore, it is necessary to propose an innovative framework for implementing the user points clustering in the infection sample collection system.

**Author Contributions:** Conceptualization, S.K.; Methodology, S.K. and X.F.; Software, S.K.; Investigation, X.F.; Writing—original draft, S.K.; Supervision, X.F.; Project administration, X.F.; Funding acquisition, X.F. All authors have read and agreed to the published version of the manuscript.

**Funding:** This work is supported by the National Natural Science Foundation of China (NSFC) under Grant No. 61902238, the China Postdoctoral Science Foundation Grant No. 2021M692493, the Science and Technology Commission of Shanghai Municipality under Grant No. 22692194600, Shanghai Natural Science Foundation under Grant No. 23ZR1426400.

**Data Availability Statement:** Data will be made available upon request..

**Conflicts of Interest:** The authors declare no conflicts of interest.

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
