# Peer review of "Blockchain-Enabled Infection Sample Collection System Using Two-Echelon Drone-Assisted Mechanism"

_drones, doi:10.3390/drones8010014_

Round 1

Reviewer 1 Report

Comments and Suggestions for Authors

The authors should go through the following review points to improve the quality of the manuscript.

1. Abstract section should be improved.

2. Introduction section should be expanded by maintaining the proper work flow.

3. The related works section should be improved by briefly describing minimum 15-20 recent relevant works with their identified gaps in a separate table.

4. The Layers of BISC should be properly explained.

5. The authors should replace all the figures with the higher resolution figures to improve the quality of the manuscript.

6. The authors should consider 2-3 more algorithms for comparison.

7. The conclusion section should be improved.

8. The authors should mention the future scope of this work clearly.

9. The authors should go through the entire manuscript and remove all the typo/grammatical errors from it.

Comments on the Quality of English Language

should be improved

Reviewer 2 Report

Comments and Suggestions for Authors

In this paper titled 'Blockchain-enabled infection sample collection system using two-echelon drone-assisted mechanism' the authors proposed a Blockchain-enabled Infection Sample Collection. In my opinion, this is a really interesting work and directly applicable to a lot of places where medicinal facility is needed. I think this algorithm can help drone delivery in a more efficient way. I recommend this work to be published.

Reviewer 3 Report

Comments and Suggestions for Authors

A peer-reviewed article focuses on an urgent task

development of a layered system for collecting biological samples using drones.

When solving such a problem, a number of intermediate optimization problems arise, for the solution of which the authors propose several effective algorithms and then evaluate the results achieved.

The tasks set by the authors were successfully solved, which was confirmed by the corresponding calculations. Algorithms are correctly designed.

The article is extremely relevant and useful.

And, of course, this article is recommended for publication.

However, there are comments.

The authors underestimate the usefulness of their proposed solutions. They timidly introduce into the problems of building a heterogeneous system for collecting biological samples using drones.

The authors in the introduction and annotations very quickly move on to setting the task of building such a heterogeneous system. The authors in the annotation continue to use the term "security," and later mention information security.

The authors need to strengthen the task of the main objective function - the need to improve the level of biological safety while limiting the consumption of resources.

1. It is necessary to define the concept of "biological safety" and gradations of its levels.

2. The authors clearly underestimate the importance of their proposal to use blockchain technology. The role of information security in achieving a high level of biosafety should be emphasized. It is necessary to show how these two concepts (biological and information security) are connected and interact with each other.

3. Recommendation to authors for future studies: in addition to two-level clustering of the biological material collection system, its clustering by sectors isolated from each other will help improve the performance of such a system. If "vertical" echeloning provides biological safety along the "vertical," then clustering will help complement this effect along the "horizontal." In general, a high synergistic effect can be achieved.

Reviewer 4 Report

Comments and Suggestions for Authors

This paper is based on Blockchain-enabled infection sample collection system using two-echelon drone-assisted mechanism. Thus, this paper is directly related to the theme of this journal.

Overall, the paper is organized properly; the concept and future research directions are extensively explained. So, the paper is accepted after following minor changes:

1.       Problem of paper and motivation is not clear in introduction

2.       Comparison of current is not given with previous research

3.       Pseudo code of algorithm is given but comments of statements are not added which make ease for readers to understand  

4.       Paper contains few grammar mistakes which will be cooperated in final version.

5.       Only few references are added in paper, but more than 50 references so to attract readers add few latest references related to this paper, which is mentioned below

 Abdullah Ayub Khan, Reem Alkanhel, Hela Elmannai, and Sami Bourouis. "Lightweight-biov: blockchain distributed ledger technology (bdlt) for internet of vehicles (iovs)." Electronics 12, no. 3 (2023): 677.

Casella, Vittorio, Filiberto Chiabrando, Marica Franzini, and Ambrogio Maria Manzino. "Accuracy assessment of a UAV block by different software packages, processing schemes and validation strategies." ISPRS International Journal of Geo-Information 9, no. 3 (2020): 164.

Laghari, Asif Ali, Awais Khan Jumani, Rashid Ali Laghari, and Haque Nawaz. "Unmanned aerial vehicles: A review." Cognitive Robotics 3 (2023).

Aggarwal, Shubhani, and Neeraj Kumar. "Path planning techniques for unmanned aerial vehicles: A review, solutions, and challenges." Computer Communications 149 (2020): 270-299.

Comments on the Quality of English Language

only minor changes required in english
